# Gender differences in the intention to study math increase with math performance

Thomas Breda [1,2] ✉, Elyès Jouini[2,3] & Clotilde Napp [2,3]

Even though females currently outnumber males in higher education, they remain largely underrepresented in math-related fields of study, with no sign of improvement during the past decades. To better understand which students drive this underrepresentation, we use PISA 2012 data on 251,120 15-year-old students in 61 countries to analyse boys' and girls' educational intentions along the ability distribution on math assessment tests. We analyze the percentages of boys and girls intending to pursue math-related studies or careers as a function of math performance. First, we show that for both boys and girls, there is a positive and linear relation between the probability of intending to pursue math and math performance. Second, the positive relation is stronger among boys than among girls. In particular, the gender gap in student intentions to pursue math-related studies or careers is close to zero among the poorest performers in math and increases steadily with math performance. Third, as a consequence, the gender gap in math performance, to the detriment of girls, is larger among students intending to pursue math than in the general student population.

Even though females currently outnumber males in higher education, they remain underrepresented in science, technology, engineering, and mathematics (STEM) university majors and jobs. STEM is, however, a broad categorisation that includes fields in which females are not anymore underrepresented, such as molecular biology and neuroscience in the United States. Scholars have emphasised the necessity of focusing more narrowly on those STEM fields that are the most math-intensive (math, computer science, physical science, geosciences, engineering and economics[1–3]), as the underrepresentation of females in these fields remains large and has not decreased in most developed countries during the past two decades[4,5]. For example, from 2004 to 2014, the share of bachelor's degrees awarded to females in engineering and computer science in the US stagnated at ~20%, while it decreased from 46 to 43% in mathematics and statistics and from 42 to 40% in physical science[5].

This underrepresentation of females in math-intensive fields is a source of concern for two main reasons. First, it contributes to sex and gender wage inequalities in the labour market, as math-intensive jobs pay more[6–8] and are also subject to a smaller gender wage gap[9].

Second, it can lead to a shortage of workers with math-related skills at a time when the demand for such skills is increasing[10]. This is all the more true if the most talented females are those who stay away from math-intensive careers the most.

There is a large body of literature on the sex and gender gaps in the pursuit of STEM or math-related studies[11–18]. This literature, however, has not considered how the gender gap evolves with math ability. An analysis of men's and womens' intentions regarding math-related studies as a function of their math ability is important for the following three main reasons. First, as mentioned, the gender gap in math-related studies is of greater concern in terms of worker shortages and wage inequality if it increases with math ability and is high among high-ability students. Second, if the gender gap in intentions to pursue math-related studies increases with math ability, that fact may imply a gender gap in math performance among math students and an underrepresentation of women among the most able students in math. Third, in terms of policy implications, an analysis along the ability distribution permits us to better identify the part of the ability distribution where interventions should be targeted to reduce the gender

[1]Paris School of Economics, 48 Bd Jourdan, 75014 Paris, France. [2]CNRS, 3 rue Michel Ange, 75 016 Paris, France. [3]Université Paris-Dauphine, Place du Maréchal de Lattre de Tassigny, 75 116 Paris, France. ✉e-mail: thomas.breda@ens.fr

gap. We are aware of only one paper analysing the sex gap in the pursuit of math-related fields at different levels of ability[19]. With a sample of ~6000 students attending a 4-year college in the US, Cimpian et al.[19] show that a high number of males whose math performance is in the bottom third of the math performance distribution are both retained in and attracted to physics, engineering and computer science (PECS) relative to the corresponding number of females.

Using data from the 2012 Program for International Student Assessment (PISA 2012) covering 61 countries, we analyse how the intention to pursue math-related studies or careers of 15-year-old boy and girl students vary along the math ability distribution. Our choice to focus on this age (in contrast to ref. 19) has two motivations. First, it allows us to obtain valid results for virtually all students, limiting selection effects related to past specialisation or dropout. Second, university degree programme choices are shaped by earlier subject choices in high school[14,20], implying that the leaky pipeline starts during high school[15,16]. As explained in ref. 15, a large share of the gender gap in STEM enrolment at university or in college is explained by STEM courses taken in high school. This shows that looking at educational or career intentions during high school rather than later on can be important to understand gender segregation across fields of study or careers. We do so for intentions to pursue math-related studies or careers specifically. We show that the positive relation between the intention to pursue math studies and math performance is stronger among boys than among girls. While the gender gap in student intentions to study math is close to zero among the lowest-ability students (1st ventile of performance), it gradually becomes larger with math ability and is highest among the highest-ability students (20th ventile of performance).

## Results
### Measuring student intentions to pursue math
Our analyses are based on data from PISA 2012, an international assessment of the knowledge and skills of 15-year-old students in mathematics, reading, and science conducted every 3 years. PISA 2012 was conducted in 2012 in the 34 (mostly) developed countries belonging to the Organisation for Economic Cooperation and Development (OECD) and an additional 30 developing countries, accounting for a total of 80% of the world economy. PISA 2012 focuses primarily on math and includes several measures of students' attitudes towards math (see Supplementary Methods for details).

PISA 2012 includes questions related to the intentions to pursue studies and careers related to math rather than to other school subjects. These intentions are measured through a series of five questions that ask students if they intend to (i) take additional math versus English/reading courses after school finishes, (ii) take a major requiring math skills versus a major requiring science skills in college, (iii) study harder than required in math versus English/reading courses, (iv) take a maximum number of math versus science classes and (v) pursue a career that involves a lot of math versus science. Our main measure of math intentions is a binary variable equal to one when the answers to the five questions above are positive. It identifies students with strong intentions to pursue math in the sense that they favour math over both reading/literature and other sciences. These students are therefore the most likely to choose math in their future studies. Below, in robustness analyses, we will show that our conclusions are robust to different constructions of the math intentions variable and, in particular, to measures that more smoothly capture student intentions.

We build our main measure for 251,120 students in 32 OECD countries and 29 non-OECD countries. A total of 23.2% of these students have strong intentions to pursue math (26.9% of the boys and 19.7% of the girls; see Supplementary Table 1). This percentage is slightly higher among the non-OECD countries than among the OECD countries (24.5% versus 22.0%), but the gender gap is smaller (5.2% versus 9%).

Our main variable captures student intentions to pursue math-related studies rather than actual enrolment. In addition, we use a dataset containing information on both student intentions and actual choices in France to show that in this country in particular, intentions and enrolment are correlated and that our conclusions hold when using actual enrolment instead of intentions. We also use data from the High School Longitudinal Study (HSLS:09) in the United States as well as previous work relying on longitudinal datasets in Canada[15] and Australia[21] to suggest that our conclusions extend to actual enrolment. Finally, we use data on adult math skills and occupations to further suggest that our conclusions may also extend to math-related or STEM occupations.

### Intentions of boy and girl students along the ability distribution
Figure 1a presents the percentages of boys, girls and all students strongly intending to pursue math-related studies or careers as a function of the ventiles of math performance standardised by country (see "Methods"). Three main features can be highlighted. First, among both boys and girls, there is a positive relation between the probability of intending to pursue math and math performance. Second, among both boys and girls, the relationship between math performance and the probability of intending to pursue math is linear. Third, the positive relation is stronger among boys than among girls. The probability of strongly intending to pursue math varies from just below 0.15 to 0.25 for girls and from just above 0.15 to 0.35 for boys. Note that this is less the case at the very high end of the performance distribution (top 5%), where girls' intentions to pursue math increase at a higher rate than at other levels of performance. A fourth stylised fact complements those observed in Fig. 1a: math performance has greater explanatory power over intentions to pursue math among boys than among girls. Indeed, the R-squared from a simple regression of strong intentions to pursue math on math performance is more than twice as large for boys as for girls (0.015 against 0.006, see Supplementary Table 1).

The difference between the percentage of boys and the percentage of girls with strong intentions to pursue math increases with math performance, as illustrated in Fig. 1b. The gender gap is very limited (1.8 percentage points) and not significantly different from zero (z-statistics = −0.89, $P = 0.375$, 95% CI = [−0.058, 0.022]) for students with the lowest levels of math ability, but it increases steadily and linearly with math ability. The gap at the top of the math performance distribution (8.0 percentage points) is more than four times larger than the corresponding gap at the bottom of the performance distribution. In Table 1, we fit linear probability models (see ref. 22 and "Methods") of the following form:

$$Math\_intention_i = \alpha Girl_i + \beta Math\_ability_i + \gamma (Math\_ability_i * Girl_i) + \delta X_i + \varepsilon_i \tag{1}$$

Equation (1) captures how the probability of strongly intending to pursue math varies with math ability for boys (coefficient $\beta$) and for girls ($\beta + \gamma$), assuming that the relationship is linear for both genders, as observed in Fig. 1a. Table 1, column 1 presents the estimates for Eq. (1) with no control variables $X_i$ and shows that a one-standard deviation increase in math performance increases the probability of boys intending to pursue math by 5.4 percentage points (z-statistics = 15.86, $P < 0.001$, 95% CI = [0.047, 0.061]) but the same probability for girls by only 3.3 percentage points. The slope for girls is approximately 40% smaller than that for boys, with the difference in slope ($\gamma = -0.021$) being statistically significant (z-statistics = −4.84, $P < 0.001$, 95% CI = [−0.029, −0.012]). The conclusions are similar when including country-fixed effects as controls and/or using alternative models such as a logit or a probit model (see Supplementary Table 3 and "Methods").

In Supplementary Fig. 1, we show that the patterns observed in Fig. 1 hold in both the OECD and the non-OECD countries and for a small sample of countries: Brazil, France, Russia, and the US. Supplementary Table 4 provides further estimates of Eq. (1) without controls for all countries in our sample. In 3 of the 32 available OECD countries and 6 of the 29 non-OECD countries, the intention to pursue math does not increase significantly with math ability (i.e., the coefficient $\beta$ in the regression $Math\_intention_i = \beta Math\_ability_i + \varepsilon_i$ applied to both girls and boys together is not significant at the 5% level; see last column of Supplementary Table 4). We therefore assume that we lack the statistical power needed to compare patterns in the choices of girls and boys in those countries (alternatively math intentions are truly not related to math ability, and it is pointless to pursue the exercise). Keeping only the other countries, we find that the positive relationship between math performance and the percentage of students strongly intending to pursue math is larger among boys than among girls in 24 of the 29 remaining OECD countries and in 20 of the 23 remaining non-OECD countries. The corresponding differences in slope are statistically significant at the 5% level in 14 OECD countries and 7 non-OECD countries. Further investigation shows no statistically significant correlation between the magnitude of the differential pattern of intentions by gender according to ability in each country (as measured in Supplementary Table 4) and countries' development ($r = 0.19$, t-statistics(55) = 1.47, $P = 0.15$, 95% CI = [−0.07, 0.43] with Gross Domestic Product or $r = 0.11$, t-statistics(53) = 0.95, $P = 0.40$, 95% CI = [−0.15, 0.36] with the Human Development Index) or countries' extent of

gender inequality ($r = 0.13$, t-statistics(54) = 0.84, $P = 0.35$, 95% CI = [−0.14, 0.38] with the Gender Gap Index).

We can also study the ratio of the percentage of boys strongly intending to pursue math to the corresponding percentage of girls (instead of the difference between the two percentages). Figure 2 is the analogue of Fig. 1b for this alternative measure of the gender gap in intentions. It presents this boys-to-girls ratio as a function of the ventiles of math performance standardised by country. We observe that this ratio does indeed increase with math performance. The ratio is close to 1 among poor performers and reaches 1.5 among high performers. This is also illustrated in Supplementary Table 5, which shows that the ratio between the probabilities of boys and girls strongly intending to pursue math increases from 1.18 in the bottom quartile of the math performance distribution to 1.4 in the top quartile.

In summary, relative to boys, high-performing girls intend less to pursue math than low-performing girls.

## Implications for girls' representation among high performers

The patterns highlighted above have implications both for the relative math performance of girls and for their representation among high performers in the subpopulation of students who intend to pursue math.

Table 2 compares the gender gaps in average math performance for the whole population and among students with strong intentions to pursue math for a selection of regions and countries. Worldwide, the gender gap (girls minus boys) in math performance is negative and ~40% larger in magnitude among students with strong intentions to

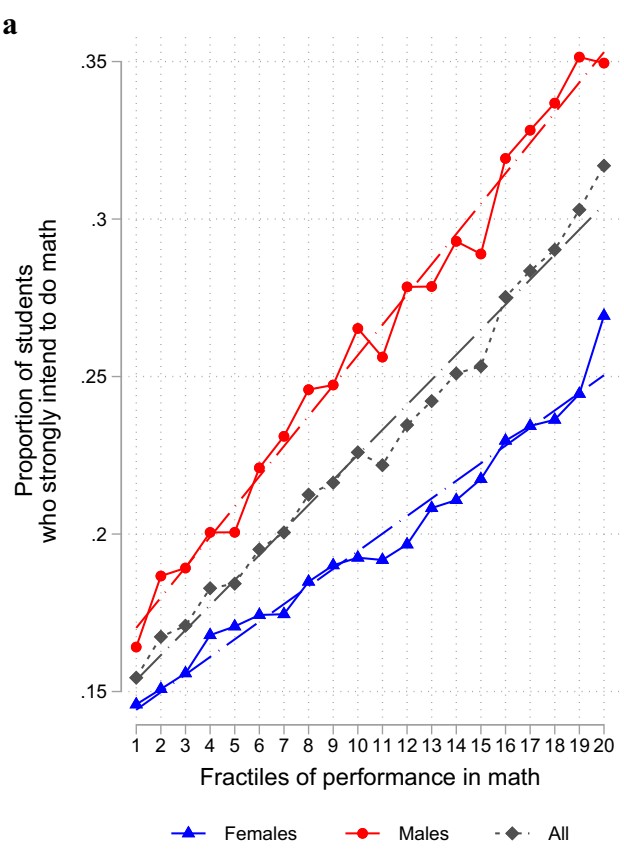

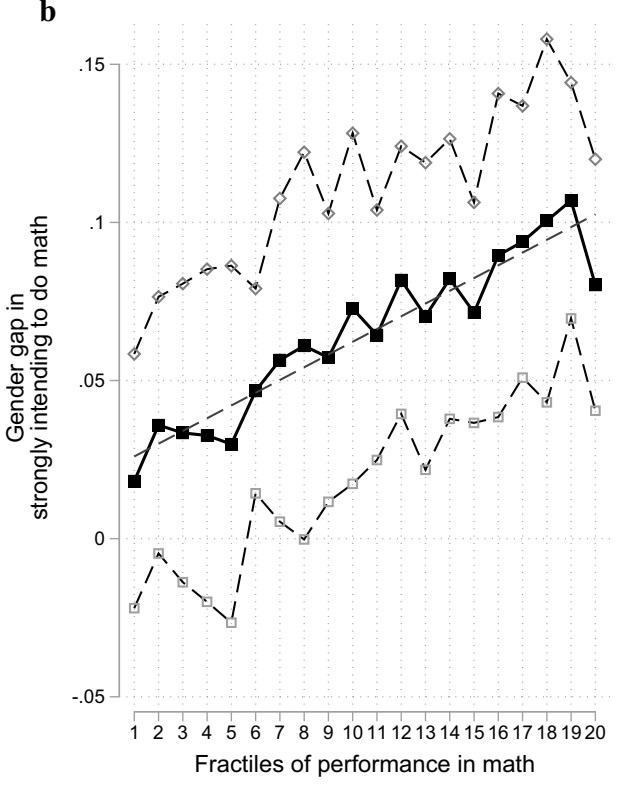

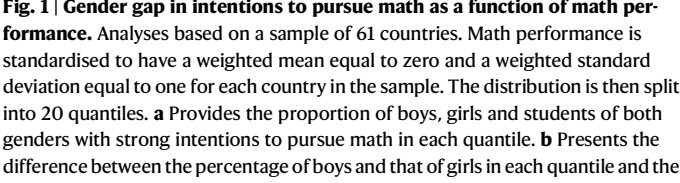

**Fig. 1 | Gender gap in intentions to pursue math as a function of math performance.** Analyses based on a sample of 61 countries. Math performance is standardised to have a weighted mean equal to zero and a weighted standard deviation equal to one for each country in the sample. The distribution is then split into 20 quantiles. **a** Provides the proportion of boys, girls and students of both genders with strong intentions to pursue math in each quantile. **b** Presents the difference between the percentage of boys and that of girls in each quantile and the

associated 95% confidence interval (i.e., mean difference +/− 1.96 SEM). A student has strong intentions to pursue math if he or she answered yes to a series of five questions capturing intentions to pursue studies or careers related to math. Estimates and standard errors are based on plausible values for math ability and account for measurement error in these values in addition to standard sampling error (see "Methods" for details).

**Table 1 | Boys and females intentions to pursue math studies (from Eq. (1)) with various sets of control variables**

| | Dependent variable is "strong intentions to pursue math" | | | | |
|---|---|---|---|---|---|
| | (1) | (2) | (3) | (4) | (5) |
| Girl | −0.0645 | −0.0345 | −0.0358 | −0.0487 | −0.0772 |
| | [−0.0718, −0.0571] | [−0.0432, −0.0258] | [−0.0468, −0.0248] | [−0.0562, _0.0413] | [−0.0852, −0.0691] |
| | P <0.001 | P <0.001 | P <0.001 | P < 0.001 | P < 0.001 |
| Math performance | 0.0541 | 0.190 | 0.0281 | 0.0433 | 0.0599 |
| | [0.0474, 0.0608] | [0.172, 0.209] | [0.0187,0.0374] | [0.0369, −0.0497] | [0.0531, 0.0668] |
| | P <0.001 | P <0.001 | P < 0.001 | P < 0.001 | P < 0.001 |
| **Girl\*(math performance)** | **−0.0210** | **−0.0424** | **−0.0212** | **−0.0231** | **−0.0116** |
| | [−0.0295, −0.0125] | [−0.0623, −0.0225] | [−0.0345, −0.00787] | [−0.0314, −0.0147] | [−0.0207, −0.00249] |
| | P <0.001 | P <0.001 | P = 0.002 | P < 0.001 | P = 0.013 |
| Other controls | None | Reading, science and math ability (and interactions with gender) | Math self-concept (and interactions with gender) | Interest for math (and interactions with gender) | Socioeconomic background (and interactions with gender) |
| Observations | 251,120 | 251,120 | 123,712 | 249,991 | 247,853 |
| R-squared | 0.0183 | 0.0380 | 0.0737 | 0.0871 | 0.025 |

Notes: Analyses based on a sample of 61 countries. Math performance is standardised to have a weighted mean equal to 0 and a weighted standard deviation equal to 1 for each country in the sample (and in the full sample of countries). "Strong intentions to pursue math" corresponds to answering yes to five questions measuring intentions to pursue a career in a math-related field or to invest more in math than in other science subjects or in reading/literature among PISA 2012 students. The rich set of controls in column 2 includes a third-order polynomial in reading ability, a third-order polynomial in science ability, reading ability interacted with gender, science ability interacted with gender, and interaction terms among math, science and reading ability (3 two-way interactions and one three-way interaction). All estimates and standard errors are based on plausible values for math ability and account for measurement error in these values in addition to standard sampling error. 95% confidence intervals in brackets. P values based on two-sided t tests.

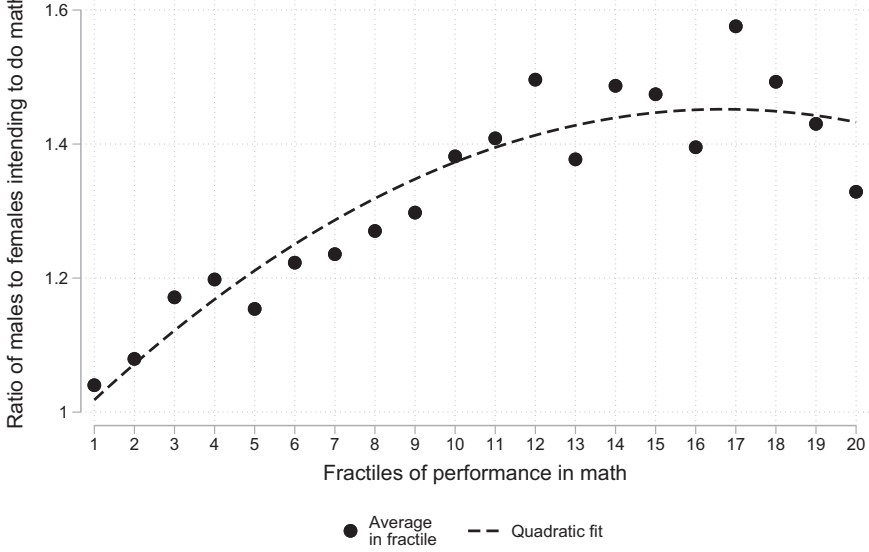

**Fig. 2 | Ratio of boys to girls with strong intentions to pursue math in each quantile of the math performance distribution.** Analyses based on a sample of 61 countries. Math performance is standardised to have a weighted mean equal to zero and a weighted standard deviation equal to one for each country in the sample. The distribution is then split into 20 quantiles. The figure provides the ratio between the percentage of boys and the percentage of girls with strong intentions to pursue math in each quantile. A student has strong intentions to pursue math if he or she answered yes to a series of five questions capturing intentions to pursue studies or careers related to math. Estimates are based on plausible values for math ability (see "Methods" for details).

pursue math than in the general population (−19.0% vs. −13.7% of a standard deviation of the math ability distribution). This difference is larger in non-OECD countries (where the gender gap in math performance is approximately 70% larger among students intending to pursue math) than in OECD countries (where it is only 25% larger). This general pattern is, however, not found systematically in all countries: the gender gap in math performance is larger among students who intend to pursue math in 22 out of 32 OECD countries and 18 out of 29 non-OECD countries (Supplementary Table 6). Note that these results

are about relative performance gaps; they have no direct implication for the absolute math performance of girls or boys that intend to do the math.

The last two columns of Table 2 and Supplementary Table 6 and Supplementary Fig. 2 also provide statistics on the representation of girls among high performers in the whole population and in the subpopulation of students who intend to pursue math. We observe that in all 61 countries except the UK and Indonesia, the ratio of girls to boys in the top decile of math performers is lower among students who

**Table 2 | Comparison of gender gaps in math performance among students who intend to pursue studies related to math and in the whole population**

| | (a) Gender gap (females minus males) in the whole population | (b) Gender gap among those strongly intending to do math | Ratio of Gender gaps (a/b) | Girls-to-boys ratio in top decile of whole population | Girls-to-boys ratio in top decile among those strongly intending to do math |
|---|---|---|---|---|---|
| All PISA 2012 Countries | −0.137 | −0.190 | 1.389 | 0.743 | 0.536 |
| | [−0.157, −0.116] | [−0.230, −0.151] | | | |
| | P < 0.001 | P < 0.001 | | | |
| OECD countries | −0.158 | −0.199 | 1.257 | 0.671 | 0.482 |
| | [−0.186, −0.131] | [−0.260, −0.138] | | | |
| | P < 0.001 | P < 0.001 | | | |
| Non-OECD countries only | −0.113 | −0.191 | 1.685 | 0.832 | 0.596 |
| | [−0.146, −0.081] | [−0.240, −0.141] | | | |
| | P < 0.001 | P < 0.001 | | | |
| **Selected countries** | | | | | |
| Brazil | −0.194 | −0.272 | 1.399 | 0.732 | 0.402 |
| | [−0.253, −0.136] | [−0.383, −0.162] | | | |
| | P < 0.001 | P < 0.001 | | | |
| France | −0.206 | −0.281 | 1.361 | 0.608 | 0.321 |
| | [−0.296, −0.116] | [−0.504, −0.057] | | | |
| | P < 0.001 | P = 0.014 | | | |
| Japan | −0.230 | −0.260 | 1.133 | 0.470 | 0.311 |
| | [−0.321, −0.139] | [−0.427, −0.094] | | | |
| | P < 0.001 | P = 0.002 | | | |
| New Zealand | −0.177 | −0.465 | 2.619 | 0.605 | 0.219 |
| | [−0.274, −0.081] | [−0.710, −0.219] | | | |
| | P < 0.001 | P < 0.001 | | | |
| Poland | −0.115 | −0.108 | 0.944 | 0.736 | 0.400 |
| | [−0.215, −0.015] | [−0.275, 0.059] | | | |
| | P = 0.024 | P = 0.205 | | | |
| Russia | −0.017 | −0.097 | 5.896 | 1.040 | 0.625 |
| | [−0.085, 0.052] | [−0.254, 0.059] | | | |
| | P = 0.638 | P = 0.223 | | | |
| Spain | −0.275 | −0.438 | 1.593 | 0.457 | 0.214 |
| | [−0.331, −0.219] | [−0.569, −0.306] | | | |
| | P < 0.001 | P < .001 | | | |
| Sweden | −0.053 | −0.079 | 1.480 | 0.780 | 0.469 |
| | [−0.144, 0.037] | [−0.213, 0.055] | | | |
| | P = 0.248 | P = 0.247 | | | |
| Tunisia | −0.262 | −0.196 | 0.750 | 0.750 | 0.715 |
| | [−0.343, −0.181] | [−0.357, −0.035] | | | |
| | P < 0.001 | P = 0.017 | | | |
| United States | −0.109 | −0.191 | 1.756 | 0.787 | 0.582 |
| | [−0.182, −0.035] | [−0.359, −0.022] | | | |
| | P = 0.004 | P = 0.027 | | | |

Notes: 95% Confidence intervals in brackets. P values based on two-sided t tests. Math ability is standardised to have a weighted mean equal to 0 and a weighted standard deviation equal to 1 for each country. Gender gaps are therefore expressed as a fraction of the standard deviation of math ability among the whole population. Those strongly intending to do math correspond to students who answered yes to a series of five questions capturing intentions to pursue studies or careers related to math. The last two columns present the ratio between the number of girls and boys in the top decile of the ability distribution both in the whole population and among those strongly intending to do math. All estimates and standard errors are based on plausible values for math ability and account for measurement error in these values in addition to standard sampling error.

intend to pursue math than in the general student population (Supplementary Table 6). Supplementary Fig. 2 also shows that among students intending to pursue math, the girls-to-boys ratio above a certain level of math performance is low and decreases sharply with level of performance, from ~0.8 when all students who intend to pursue math are included to 0.5 when we focus only on the top ventile of students who intend to pursue math.

Note that these patterns may be attributable to factors other than the variations in girls' and boys' intentions to pursue math as a function of math ability depicted in Fig. 1a. First, they may be partly attributable to differences in the ability distribution (in the general population) between girls and boys, which, on their own, may imply that the gender gap in performance is larger among students intending to pursue math, even if girls' and boys' intentions to pursue math vary similarly with their ability. Second, the low representation of girls among the high math performers intending to pursue math might be attributable to our observation that in the general population, girls intend less to pursue math than boys.

In the Supplementary Note, we show that the difference in the intention to pursue math between girls and boys as a function of their performance in itself generates a larger gender gap in math performance and a lower representation of girls at the top of the ability distribution among the population of students intending to pursue math than in the general population. First, we use a reweighting procedure to equate the weighted math ability distributions for girls and for boys in the general population, and then we recompare the gender differences in math ability in the general population and among students intending to pursue math using the corresponding student weights. This procedure guarantees that the comparisons are no longer driven either by the percentage of girls and boys intending to pursue math or by the initial gender differences in the math ability distribution. We still observe that the gender gap in math performance is higher (but to a lesser extent) among students intending to pursue math than in the general population and that the ratio of girls to boys decreases with math ability (but to a lesser extent) in the former group while it is by construction constant in the latter (Supplementary Table 7).

More generally, we demonstrate in the Supplementary Note the following theoretical property: if boys and girls have the same ability distributions and if the ratio between the percentage of boys and the percentage of girls intending to pursue math increases with math ability, then, among students who intend to study math (i) the average math performance of boys is higher than that of girls and (ii) the ratio of girls to boys above a given level of performance decreases with that level. We have seen above that the condition on the boys-to-girls ratio intending to pursue math is satisfied by our data (see Fig. 2). Hence, it follows from the theoretical property above that if the students had the same initial ability distributions, the difference between girls and boys in their intention to pursue math as a function of their math performance would by itself lead to both a lower average relative performance of girls than in the general population and a decline in the representation of girls when performance increases among students who intend to pursue math.

## Robustness checks

To support our conclusions, we run a number of robustness checks. First, our results still hold when alternative measures of math performance are used. They hold across the included countries and are even stronger when we consider a measure of math performance that is not standardised by country (Supplementary Fig. 3 and Supplementary Table 8, column 2). A possible explanation is that more developed countries have higher levels of math performance and larger gender gaps in math intentions (e.g., ref. 23). Our results also hold if we consider the math performance quantiles of the gender-specific student

performance distributions instead of quantiles of the whole math performance distribution (Supplementary Fig. 4).

Second, our results are not entirely driven by our definition of intentions to pursue math-related studies or careers. Indeed, they remain valid when we instead consider an index that more continuously captures student intentions to pursue math-related studies or careers (Supplementary Table 8, columns 3 and 4). The results are also robust to the use of dummy variables built using various cut-offs within this index rather than the highest value, which corresponds to students responding yes to all underlying questions (Supplementary Table 8, columns 5 and 6).

### Actual enrollment in math-related studies/careers vs math ability

We now present evidence to assess the extent to which the patterns observed for educational and career intentions in high school may extend (i) to actual enrolment in math-related education and, later on, (ii) to careers in math-related jobs.

Regarding enrollment in math-related studies, we start by using a data source for France that contains information on ~12,000 10th-grade students and 5500 12th-grade students enrolled in French high schools in the Paris region and observed during the 2016–2017 and 2017–2018 academic years. The data were collected for another research project, and detailed information on them can be found in ref. 24. The dataset includes information about sex and math ability measures in the form of national exam scores, with the exams taking place at the end of 9th grade (Diplôme National du Brevet des Collèges) for 10th graders and at the end of 12th grade for 12th graders (Baccalauréat scientifique). For the 10th graders, the dataset also includes information on intentions to specialise in a scientific track (Première S, emphasising math, physics and natural science) in 11th grade, which is declared in 10th grade between January and March, as well as actual enrolment in the scientific track in grade 11. For 12th graders, it includes intentions to enrol in a math-related programme after high school and detailed information on actual enrolment after high school. From the observed higher education enrolments, we construct indicator variables for enrolment in a STEM programme, in a math-related programme (math, physics, engineering or computer science), and in a selective (prep school) math-related programme. These variables are similar to those used in ref. 24, and more details can be found there. We have excluded repeaters, as intentions to repeat are not collected, implying that repeaters' intentions cannot be equivalent to their actual enrolment.

Overall, the correlations between study intentions and actual enrolment are not perfect, but they are strong. For 10th graders, the correlation between their intentions to enrol in a scientific track in 11th grade as measured in 2016–2017 and their actual enrolment in a scientific track in the academic year 2017–2018 is 0.78 (t-statistics(12885) = 143.01, $P < 0.001$, 95% CI = [0.777, 0.790]). For 12th graders, the correlation between their intentions to enrol in a math-related major after high school as measured in 2016–2017 and their actual enrolment in such a major in higher education in the academic year 2017–2018 is 0.74 (t-statistics(3159) = 61.55, $P < 0.001$, 95% CI = [0.722, 0.754]). These results show that intentions are a reasonably good proxy for students' actual educational choices, at least during high school and at college entry.

Turning to the main results, we first consider a general survey question asking both 10th and 12th graders if they would consider pursuing math, physics or computer science after high school and examine how responses vary with math performance and sex using Eq. 1 above. We observe that as (standardised) math performance increases, this is more often the case, but to a lesser extent for females than for males ($\gamma = -0.023$, t-statistics(16520) = −3.38, $P < 0.001$, 95% CI = [−0.037, −0.010], see Supplementary Table 9, column 1). We also study the intentions of 10th graders (when they are in 10th grade) to

enrol in a scientific track in the following year as well as their actual enrolment in such a track (in the following year, i.e., when they are in 11th grade). If we focus on actual enrolment in a scientific track, the results align with those from our main analyses (columns 2 and 3). Note, however, that these findings concern science education in general rather than math-related fields only, as we have no information on the latter in grade 10. We can, however, use the rich survey and administrative data that we have on 12th graders' educational choices. For these students, whether we look at their intentions to enrol in a math-related major or their actual enrolment in such a major, we find that these outcomes increase with their math performance but to a lesser extent for females than for males (Supplementary Table 9, columns 4 and 5 and Supplementary Fig. 5). We finally find that these results still hold if we focus on enrolment in a STEM programme after high school (column 6) or, more narrowly, in selective math-related higher education programmes only (column 7). This last result shows that our conclusions for France are still found when we focus on the most demanding and prestigious math-related higher education programmes. Therefore, they are not exclusively driven by females being overrepresented among students whose math performance is in the bottom of the math performance distribution and who enrol in educational programmes that are not demanding in terms of math content (and may potentially fall outside the realm of STEM or PECS studies). Interestingly, when focusing on the probability of enrolling in a math-related programme after high school, we observe that this probability increases with math ability twice as slowly for females than for males (the slope for females is about half the size of the slope for males). This difference in slopes between females and males is similar to that observed worldwide on PISA regarding intentions to pursue math at 15 years old. Together, these results show that in France, our conclusions regarding the intention to study math also extend to actual enrolment in math-related studies (and even STEM studies), at least at college entry.

We also use a dataset from the High School Longitudinal Study (HSLS:09), a nationally representative US study conducted by the National Center for Education Statistics (NCES), designed with a focus on STEM. We show that our results hold when we consider the actual enrolment of high school students in a given set of courses in engineering, computer science and math as an outcome depending on their sex (see Supplementary Note for more details). These outcomes are particularly important since the leaky pipeline starts in high school level and high school choices impact college choices (see ref. 15). Approximately 17% of males but only 9% of females enrol in these courses. As illustrated in Supplementary Fig. 6, the percentage of males enrolling in these courses increases much more with math performance than that of females (from 7% to 25% for males and from approximately 7% to 10% for females). As a consequence, the sex gap in the percentage of males and females choosing these courses increases with math ability from 1.0 percentage point among the first decile of the math performance distribution (t-statistics(16283) = −0.59, $P = 0.55$, 95% CI = [−0.044, 0.023]) to 17.9 percentage points among the top decile (t-statistics(16283) = −10.74, $P < 0.001$, 95% CI = [−0.212, −0.146], see Supplementary Fig. 7). The ratio of males to females enrolling in these courses also increases with math ability from 1.14 in the bottom decile to 2.91 in the top decile of the math performance distribution (Supplementary Fig. 8). Finally, the sex gap in math performance is close to zero (0.3% of a standard deviation) and not statistically different from zero among all high schoolers (t-statistics(17115) = 0.21, $P = 0.834$, 95% CI = [−0.026, 0.032]), but it amounts to 25.0% of the standard deviation of the math ability distribution among those who choose these courses (t-statistics(2352) = −6.15, $P < 0.001$, 95% CI = [−0.329, −0.170]), clearly confirming our results based on intentions reported in PISA.

To investigate whether our results may also extend to sex gaps in careers, we examine how the probability of working in a math-related

occupation varies with math ability for male and female workers. To do this, we use data from the Programme for the International Assessment of Adult Competencies (PIAAC), a programme for the assessment and analysis of adult competencies. PIAAC measures adults' proficiency in key information-processing skills (literacy, numeracy and problem-solving) and provides information on labour market outcomes, including detailed occupation information and the usage of numeracy skills at work, for a large representative sample of approximately 150,000 working-age adults in 31 (mostly developed) countries (see details in Supplementary Methods).

Math-related jobs are identified using the 4-digit level International Standard Classification of Occupations (ISCO) available in PIAAC. The probability of working in such a job increases with math ability for working-age males 86% more than for their female counterparts (Supplementary Fig. 9a). Interestingly, this pattern is also observed when we focus more broadly on whether the respondent works in a STEM job (Supplementary Fig. 9b). Second, based on worker responses to a set of survey questions on the usage of numeracy skills at work, we observe that the sex gap in math usage at work also increases with math ability (Supplementary Fig. 9c, d). Hence, regardless of whether we measure adults' math usage at work from a direct survey question or based on their official ISCO occupation, we continue to observe the same pattern as in PISA. Note that our results are not driven by our particular classification of jobs into math-intensive (STEM) and non-math-intensive (non-STEM) jobs. Accountants, for example, are never categorised as STEM but are included in math-related occupations in the robustness checks (see details in Supplementary Methods).

Various interpretations might be possible for the results observed within the adult population. For example, adults' math ability itself depends on the amount of math required by their jobs, and if the amount is larger for male workers than for female workers, even within math-related occupations, we could obtain the pattern observed in Supplementary Fig. 9. However, these results also suggest that the patterns observed in students' educational intentions at 15 years old might have long-term consequences and still be observed in the labour market. In all cases, the results for adults do imply that the sex gap in math performance is 15–20% larger among workers in math-related occupations or workers using math intensively at work than it is in the general adult population. Females are also highly underrepresented among top math performers in these occupations: their share in the top decile of math performance drops from 38% in the general adult population to 20% among math practitioners (Supplementary Table 10). Hence, among adults who use math in their careers, the sex gap in math performance is greater.

### Examination of possible explanations

The literature has provided possible explanations for the gender gap in the choice of math studies, including gender differences in comparative advantages for math versus reading[13] and in math-related self-concepts or interest in math[3,25,26]. We examine whether these factors could account for the positive association between math performance and the gender gap in the probability of intending to pursue math.

Girl students who are good at math are more likely than their boys counterparts to be even better in reading, and this comparative advantage in reading over boys among girls can account for a large part of the raw gender gap in the intention to pursue math[11–13,25]. Similarly, student performance in reading and comparative advantages in reading over math are likely to vary with math ability and may vary differentially for girls and boys. As student intentions to pursue math are in part driven by a trade-off between their math performance and their performance in other subjects, our main results could potentially be explained by the joint distribution of performance in math, reading and science for girl and boy students. We show that this is not the case. First, the gender gap (boys minus girls) in the

comparative advantage in math (defined as math ability minus reading ability) decreases with math ability (Supplementary Fig. 10a), implying that, if anything, this comparative advantage should lead to a lower gender gap in the intention to pursue math among high achievers in math than among low achievers. Second, when we include a rich set of control variables that capture reading and science ability as well as their interaction with gender in Eq. (1), the difference between girls and boys in the relationship linking math ability to math intentions is actually magnified (Table 1, column 2). Third, even when we allow the comparative advantage in math to interact with math intentions differentially by both gender and math ability (by including math ability, gender and comparative advantage as well as the 3 corresponding two-way interactions and the three-way interaction in a regression model), the coefficient of interest capturing how math ability interacts with math intentions differentially for girls is barely affected (Supplementary Table 11). Fourth, this coefficient remains unaffected if we allow comparative advantages in math to influence math intentions non-linearly. We do so by controlling either for a dummy variable indicating a positive comparative advantage or by including comparative advantage quintiles and their interactions with student gender (Supplementary Table 11).

Gender differences in math self-concepts (i.e., how students perceive their own math ability) have often been considered in the literature as an explanation for the gender gap in math enrolment (e.g. refs. 3, 25, 26). A series of questions in PISA 2012 makes it possible to build an index to measure math-related self-concepts at the student level. Boys' and girls' self-concepts increase with math performance, but interestingly, math-related self-concepts increase more for girls than for boys, and gender differences in the way that students perceive their math ability decrease with math performance (Supplementary Fig. 10b). The gap at the top of the math performance distribution is approximately half the size of the corresponding gap at the bottom end, suggesting that math-related self-concepts cannot account for our results. A linear regression that controls for student self-concept and its interaction with student gender confirms this idea more formally (Table 1, column 3).

Similar results are obtained when we replace our measure of math-related self-concept with students' declared interest in math, which can be considered a measure of their preferences. We also find for this variable that the evolution of the gender gap along the ability distribution is opposite to that of the gender gap in intentions to do math (Supplementary Fig. 11); hence, declared preferences do not seem to account for the differential pattern of intentions to pursue math (Table 1, column 4).

Finally, we find that the gender gap in the intention to pursue math is actually larger among high socioeconomic status (SES) students. Controlling for SES differentially by gender in Eq. (1) actually lowers the magnitude of $\gamma$ by 45% (from −0.0210 to −0.0116, see Table 1, column 5). This shows that our main results partly capture the fact that high math performers tend to predominantly have a high SES.

## Discussion

We have shown that the gender gap in intentions to pursue math increases with students' math performance. Relative to their boy's counterparts, high-performing girls intend less to pursue math than low-performing girls. This pattern implies that among students intending to pursue math, gender gaps in math performance get larger than in the general population. It cannot be explained by gender gaps in comparative advantage for reading relative to math, nor by gender gaps in math self-concept or interest for math. It is, however partly explained by students' SES. In this section, we discuss some limitations of our analysis as well as how these results fit into the literature.

A first limitation is that we cannot provide evidence for a clear mechanism that explains our results. There are indeed various factors that could be behind the differential relationship between math

performance and boys' and girls' intentions to pursue math. One of these factors is SES that can explain about half of the differential relationship. This could be for several reasons. First, it could be because high-income parents spend more time and money on their children and invest in more gendered activities[27]. Hence, they may transmit gender norms, including those related to math careers[28], earlier and to a greater extent[29]. At a more macro level, gender gaps in math education are known to be larger in socioeconomically advantaged school districts[29] and in richer countries[30,31]. The type of educational environment that is likely to surround high math performers within a country or to prevail in more privileged school districts or in more developed countries in general typically implies more difficult curricula, higher performance standards, and greater competition. Such contexts may affect differentially girls and boys. They have in particular been shown to heighten gender-essentialist ideas about math and science[32]. Hence, both parental background and school environment may explain why the salience of general gender stereotypes regarding math could increase with math performance, leading to a larger gender gap at higher levels of performance. Parental background can however also affect students through other and possibly more direct channels. For example, high- and low-SES parents have different occupations on average, and perhaps the gender divide between parents' occupations is higher among high-SES students, leading them to make more gendered educational choices. Further research would be necessary to understand exactly why SES can explain about half of the differential relation between girls' and boys' intentions to pursue math and their math ability.

In all cases, SES can only explain half of our main result. What could drive the unexplained part? Although we are not able to provide supporting evidence, we ultimately suggest that gender stereotypes could provide a plausible general explanation. Gender stereotypes relating math primarily with boys have been considered in the literature as explanations for gender differences in choices of math and science studies and careers[23,33,34]. They could possibly account for the differential relation between intention to pursue math and math performance among girls and boys, for example, because they would be more salient to students who perform well in math. Indeed, these stereotypes can incorporate the idea that girls cannot excel at math, a discipline where it is often believed that innate talent is needed. The existence of the stereotype that girls cannot excel at math has been documented in the literature[2,35,36]. This could imply that when facing different possible curricula, high-performing girls may stay away from math and assume that they have a better chance to realise their full potential if they specialise in another field. This interpretation is also supported by several papers that have documented that "it is often the most talented members of stereotyped groups that are most affected by others' biased perceptions and, more generally, by signals suggesting that they may lack ability" ([37]; see also refs. [38–40]). Note that gender stereotypes would then generate a self-fulfilling mechanism: the stereotype that girls cannot be excellent at math would dissuade the most talented girls from studying math, leading girls to have lower math performance than boys among math students, which would contribute to maintaining the initial stereotype.

Another potential limitation of our analysis is that our main variable of interest captures intended rather than actual student behaviours. This allows us to provide consistent and directly comparable evidence across a large set of countries having very different educational systems. However, it is not entirely clear that intentions do indeed materialise into actual choices, nor that the corresponding choices lead to careers that are considered PECS or STEM. Students who intend to pursue math-related studies may opt for math-related jobs that would not be counted as a STEM career as, for example, teaching math or accounting. To address this limitation, we retrieved country-level data collected by the UNESCO on the percentage of graduate students who graduate in a STEM field, and we show that these percentages are broadly comparable in terms of magnitude to the country-level averages for our binary measure of math intentions (Supplementary Table 1). More importantly, we have shown that intentions and actual choices are correlated in one country (France) and that our conclusions extend to actual enrolment in France and the US. However, for other countries, we cannot entirely affirm that our observations regarding intentions apply to actual enrolment.

Note that, a number of previous papers on secondary and post-secondary course and degree field choices, although not focusing on math ability and having different objectives, make observations or present results that are in line with ours.[25], relying on administrative data on a full cohort of students in Australia, obtain results on the choice of physics, information technology, and advanced math that are in line with our conclusions (see Fig. 2 in ref. [25]).[15], relying on the Educational Longitudinal Study of 2002 (ELS:02) in the US, observe that "females who pursue PEMC [physics, engineering, mathematics and computer science] majors in college are not the females who were the highest performers in high school".[15], relying on Canadian data, show that "many higher achieving female students tend to select non-STEM courses in the last years of high school, despite having done well in math in 11th grade. These choices may detract from their being able to register in STEM programmes".

Our conclusions differ from those obtained in ref. [19] on a sample of ~6000 students attending a 4-year college in the US. This difference may be explained in part by differences in the samples considered in the two studies. Cimpian et al.[19] restrict their sample to students attending a 4-year college, a subsample that represents ~25% of all high schoolers and is not representative of the whole population of high schoolers, as females, especially those whose math performance is at the left side of the math performance distribution, are over-represented. Indeed, as shown in ref. [15], many females go to college to study humanities ("when they are not STEM ready", i.e., when they have not taken enough high school courses in science and math), whereas males go to college mainly to pursue scientific studies. Another possible reason for the different results obtained by Cimpian et al.[19] is related to the limitation that we mentioned above: we focus on intentions to pursue math while they focus on actual enrolment in PECS majors. However, we think that this explanation is less likely, as we have shown that our results do extend to actual enrolment in a number of countries, including the US, when (course) enrolment is measured among a representative sample of high schoolers.

In this study, we show that the gender gap in intentions to pursue math is larger among high math achievers. The intentions of girls to study math are less related to their math performance than those of boys. This phenomenon may induce lower representation of high-performing girls in math-related fields and may exacerbate the small gender gap in math performance observed in the general student population by the age of 15. These features may in turn contribute to sustaining the stereotype that math is not for girls and could contribute to explaining how the gender gap in enrolment in math-related fields can be so pervasive in the long run despite extensive policy attention and in contrast to what has happened in several prestigious professions (e.g., law or medicine).

A potential future implication of our results is in terms of public policy. Indeed, our conclusions suggest that it might be useful to design interventions that can encourage more high-performing females to enrol in math-related fields. Such interventions would place more girls among highly visible top math students and practitioners. These girls could in turn act as natural role models and induce additional math-related vocations among girls. At this stage, these implications are speculative and remain to be tested in future work.

More generally, our work emphasises the importance of analysing the impact of any policy aiming to equalise the representation of girls in math-related fields based on its impact on the choices of boys and girls along the whole ability distribution.

## Methods

### General description of PISA 2012

The Programme for International Student Assessment (PISA) is an every-three-year international survey of 15-year-old students aimed at determining their knowledge and skills in different domains. Students' abilities are assessed in the three curricular domains: mathematics, reading, and science. Students also answer a background questionnaire, seeking information about the students themselves, their homes, and their school and learning experiences. School principals also complete a questionnaire that covers the school system and the learning environment.

The assessment does not just ascertain whether students can reproduce knowledge; it also examines how well students can extrapolate from what they have learned and can apply that knowledge in unfamiliar settings, both in and outside of school.

The PISA target population is made up of all students in any educational institution between the ages of 15 years and 3 months and 16 years and 2 months at the time of the assessment. This specific age has been chosen because it is close to the end of compulsory education in most countries. Efforts have been made to ensure the absence of cultural or national biases in the test items and in the evaluation of performance.

We analyse data from the PISA 2012 survey[41,42]. The student dataset in 2012 contains around 510,000 observations, which roughly represent a population of 28 million 15-year-olds attending seventh grade or above in 64 countries, including the 34 countries belonging to the OECD in 2012.

### Data restrictions on PISA

PISA treats Florida, Connecticut and Massachusetts separately from the rest of the United States because they have a decentralised (specific) management of the PISA survey. This is also the case for the Perm region of the Russian Federation. We have integrated these states/regions to the country they belong to. PISA also considers separately Chinese provinces and cities that claim their independence from China or have strong cultural specificities: Shanghai, Macao, Taiwan and Hong Kong. We have not grouped these cities/regions in a single "China" sample as such grouping may hide strong cultural differences.

PISA surveys systematically assess students' performance and knowledge in three core subjects: mathematics, reading and science. However, one of the three core subjects is chosen to be covered in greater depth in each survey. In 2012, mathematics literacy is the major subject area, as it was in 2003. This allows us to get more in-depth information on the students' mathematics skills. On top of taking math tests, students fill out a background questionnaire that provides contextual information about themselves, their homes, and their school and learning experiences. The background questionnaire takes 30 min to complete and seeks information about students' engagement with and at school in general and engagement with mathematics in particular. It includes questions on students' motivation to succeed in mathematics, the beliefs they hold about themselves as mathematics learners, and their dispositions and behaviours in math-related fields. Of particular interest to us are questions about students' intentions to pursue math-related studies and careers, as well as questions about their self-concept, their instrumental or intrinsic motivation for math, their math behaviours, and about subjective norms in mathematics (see details below).

The student questionnaire has a rotation design, which means that all students do not answer the same set of questions. The rotated design is such that there are three different forms of the questionnaire, each containing a common part and a rotated part. The common part (which is administered to all students) contains questions about gender, language at home, migrant background, home possessions, parental occupation and education. The rotated part (which is administered to one-third of students) contains questions about attitudinal and other non-cognitive constructs. The rotation design is such that constructs are asked in two out of the three different forms of the questionnaire to allow for joint analyses of these constructs. This results in responses from two-thirds of students per construct.

Our main variable of interest capturing intentions to study math is obtained through a series of five items in a questionnaire only administered to two-thirds of PISA participating students (selected randomly). As a consequence of this design combined with non-response to at least one of the five questions by about 15% of the students, we could build this variable for 251,120 observations (out of the 510,000 ones for which math ability and gender are also observed) in 32 OECD countries and 29 non-OECD countries. All analyses involving intentions to do math are made on this sample. Only the analyses examining how girls' and boys' math attitudes vary with their math performance (and not involving math intentions at all) are made on samples including students for which math intentions may be unknown (e.g., Supplementary Fig. 11). This choice has no substantial incidence on the results.

### Variables of interest in PISA 2012

Our first variable of interest in PISA 2012 measures intentions to do math. PISA 2012 asked students to report their intentions to use mathematics in their future studies and careers. Five items (ST48) used the forced-choice format to measure students' plans regarding mathematics at some stage in the future. The first item had students decide between taking additional courses in mathematics or the language of the test after school finished. The second item asked whether students planned on majoring in a subject requiring mathematics or science at college (or equivalent educational institutions in different countries). The third item asked whether students were willing to study harder in either their mathematics classes or in classes teaching the language of the test. For the fourth item, respondents had to indicate whether they were planning on taking as many mathematics or science classes as possible during their education. The fifth and last item of that battery required respondents to choose whether they were planning on pursuing a career that involved a lot of mathematics or science. Our main measure of "intentions to do math" is a binary variable equal to one for students who choose math in all five items.

Students' responses to these five items are also used to create the index (MATINTFC), reflecting the extent to which a student intends to pursue math studies and careers. We use it in robustness analyses.

PISA 2012 also provides information about students' self-concept through their responses as to whether they strongly agree, agree, disagree or strongly disagree that (i) they are just not good in mathematics; (ii) they get good grades in mathematics; (iii) they learn mathematics quickly; (iv) they have always believed that mathematics is one of their best subjects; (v) they understand even the most complex concepts in mathematics class. Students' responses are used to create the index of mathematics self-concept which was standardised to have a mean of 0 and a standard deviation of 1 across OECD countries. Our sample includes 314,607 observations about self-concept, and since self-concept and intentions in mathematics are not in the same question set, the number of students of which we can build our main measure of math choice and a measure of math self-concept is reduced to 123,712.

PISA 2012 also assesses intrinsic motivation or declared interest for math. It refers to the drive to perform an activity purely for the joy gained from the activity itself. Students are intrinsically motivated to learn mathematics when they want to do so because they find learning mathematics interesting and enjoyable and because it gives them pleasure, not because of what they will be able to achieve upon mastering mathematical concepts and solving math problems. PISA 2012 measures students' intrinsic motivation to learn mathematics through students' responses as to whether they strongly agree, agree, disagree or strongly disagree that (i) they enjoy reading about mathematics; (ii)

they look forward to mathematics lessons; (iii) they do mathematics because they enjoy it and (iv) they are interested in the things they learn in mathematics. Students' responses are used to create the index of intrinsic motivation for math which was standardised to have a mean of 0 and a standard deviation of 1 across OECD countries. Our sample includes 316,708 observations about intrinsic motivation, and 249,991 observations about both intrinsic motivation and intentions in math.

The Programme for the International Assessment of Adult Competencies (PIAAC) is a programme of assessment and analysis of adult competencies. Countries were included in the survey in three rounds between 2011 and 2018. The Survey is implemented by interviewing adults aged 16–65 in their homes–about 5000 individuals in each participating country. Questions are usually answered via computer, although the survey can also be implemented via pencil-and-paper. The survey assesses literacy and numeracy skills as well as the ability to solve problems in technology-rich environments. It also collects a broad range of information, including detailed occupation, how skills are used at work and in other contexts, such as the home and the community. The Survey is designed to be valid cross-culturally and internationally. This means that countries are able to administer the survey in their national languages and still obtain comparable results.

The participating countries and the number of observations in each of them are provided in Supplementary Table 12.

Our final samples contain 148,023 adults aged 16–65 with a valid occupation and information on their numeracy skills and 138,630 adults aged 16–65 with information on their numeracy skills usage of these skills at work.

We first classify as STEM occupations the 2-digit ISCO occupations 21, 31, 25 and 35 as well as the 3-digit ISCO occupations 331, 431 and 241 and the 4-digit ISCO occupations 3352, 1211, 1223, 2356 and 2631. ISCO occupations 9999, 231, 232 and 233 are excluded from the analysis as it is not possible to know if they correspond to STEM jobs.

We then classify as math-intensive occupations those that are classified as STEM (see above) except the 3-digit ISCO occupations 213, 216, 315, 314, 313 and 312 and the 4-digit occupation 1223.

We have also considered alternative classifications, for example, by also considering as math-intensive occupations the ISCO 3-digit occupations 241, 231, 232 and 233 and the ISCO 4-digit occupations 1211, 2161, 2164 and 2165. Our results are robust to these alternative constructions (results available upon request).

Usage of numeracy skills at work is measured through a set of five questions capturing various ways of how those skills may be used. In the same logic as for our analyses of PISA data, we build an indicator variable equal to 1 for individuals who have answered they use numeracy skills at all five questions. We also use directly the index variable available in PIAAC capturing continuously the intensity of usage of numeracy skills at work (see Supplementary Fig. 10d).

### Weights and representativeness
PISA provides weights to make the sample of surveyed students representative of the 28 million 15-year-old students in the surveyed countries. We use these weights in all country-specific and worldwide analyses, so the results that we provide are not subject to sample selection and are consistent estimates of the underlying parameters at the country and world levels. Note that the sum of the PISA weights by country is proportional to the countries' total populations, so that more populated countries are given greater weight in the worldwide statistics.

### Standardised variables
To obtain results that can be interpreted both globally and country-by-country, we normalise math performance so that its weighted mean is zero with a weighted standard deviation of one for each country. This transformation implies that math performance has the same mean and

standard deviation in each country and makes it easy to obtain gender gaps in math performance that are directly expressed as a fraction of a standard deviation and, as such, are directly comparable across countries. The transformation is performed over the whole sample of students for whom math ability is observed. It is performed separately for each plausible value of math ability. We also perform this transformation for reading and science ability when those variables are used. Unless otherwise specified, outcome or control variables that are measured as indexes taking on several values are standardised so that their weighted mean is zero and their weighted standard deviation is one for the whole sample of countries (but not necessarily in each country). Finally, dummy variables for which gender gaps can be expressed simply as a percentage point difference in the rate of positive answers are the only variables that are never standardised.

We also provide results with alternative standardisations as robustness checks. In particular, the results based on math ability are standardised at the world level so that differences in means and standard deviations across countries are maintained.

### Statistical model
We mainly use linear probability models to analyse how intentions to pursue math-related studies or careers vary with math ability or other covariates. As logistic models are commonly used to analyse categorical dependent variables, we also verify that our main conclusions are robust to the alternative use of a logit or a probit model (Supplementary Table 3). We primarily use a linear probability model for two main reasons. First, the linear probability model makes it possible to directly interpret the regression coefficients as marginal effects, and the results are therefore easier to understand. In our case, they also make it possible to more directly link the regression results to the figures in the paper. Indeed, fitting a linear probability model is almost equivalent to fitting the slopes of the relationship between math performance and the intention to pursue math among girls and among boys (Fig. 1). Second and more importantly, a nonparametric examination of the relationship between math ability and the intentions to study math suggests that the relationship between the two is close to linear (Supplementary Fig. 12). This provides support for the use of a linear probability model (which assumes linearity) instead of other models that rely on different parametric assumptions that do not seem clearly supported by the data.

### General estimation procedure for PISA and PIAAC
Details about the PISA methodology can be found in the PISA Technical Reports[41,42], and only a basic summary is reported here. PISA and PIAAC adopt item response theory and provide only a set of plausible values for each student's actual scores. These plausible values (5 for PISA 2012, 10 for PIAAC) are random numbers drawn from the distribution of scores that could be reasonably assigned to each individual, given his or her answers–that is, the marginal posterior distribution. Any estimation procedure in PISA or PIAAC (for instance, the mean score for males) involves the calculation of the required statistic for each plausible value (appropriately weighted with the reported students' weights), and the final estimate is the arithmetic average of the five (ten) estimates obtained.

### Statistical inference
Standard errors are calculated with a replication method that takes into account the two-stage stratified sample design for the selection of schools and of students within schools[42]. The sources of uncertainty in PISA and PIAAC are actually twofold. First, as explained above, there is some uncertainty in the ability measure for each student (adult), as PISA (PIAAC) provides five (ten) plausible values drawn from a posterior ability distribution. Second, there is standard sampling error at the country level, as performance gaps are not established over the universe of students (adults) in a given country. To deal with sampling

error, PISA and PIAAC provide 80 alternative sets of individual weights and detailed guidelines on the use of those weights. The computation of the corrected standard errors basically relies on bootstrap techniques: the regression of interest must be run for each of the five plausible values, weighting the regression first with the true set of individual weights and then with the 80 alternative sets of weights. The correct point estimate is the average of the five regressions (ten regressions for PIAAC) run with the true set of weights, while the standard error is computed according to a formula that sums both of the measurement errors described above. All our results (from simple means to more complex regressions) involving individual-level measures of ability are produced using the procedure described above.

## Reporting summary

Further information on research design is available in the Nature Portfolio Reporting Summary linked to this article.

## Data availability

The main data sources used in this research come from the 2012 Programme for International Student Assessment (PISA 2012, https://www.oecd.org/pisa/pisaproducts/pisa2012database-downloadabledata.htm) and the Programme for the International Assessment of Adult Competencies (PIAAC, https://www.oecd.org/skills/piaac/publicdataandanalysis/). Both are managed by the OECD, and the data are publicly available. HSLS:09 data are publicly available at https://nces.ed.gov/surveys/hsls09/. The robustness checks for France presented in Supplementary Table 9 are based on data used in ref. 24 and whose access is restricted. To access these data, a request must be sent to the statistical institute of the French Ministry of Education (the DEEP, see ref. 24 for details).

## Code availability

All the codes used for our main analyses are publicly available and deposited at Zenodo[43]. They can be accessed at https://doi.org/10.5281/zenodo.7181225.

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

## Acknowledgements
Financial support from the Women and Science Chair (a Dauphine Foundation Chair in partnership with Amundi, Fondation L'Oréal, Generali France, La Poste, Safran and Talan) is gratefully acknowledged by all authors. We also thank Pia Busschaert for tremendous research assistance.

## Author contributions
T.B., E.J. and C.N. are all first authors. T.B., E.J. and C.N. contributed equally to the definition of the research, the writing of the paper and the empirical analyses. T.B. accessed the data from the French Ministry of Education and did the analyses for France.

## Competing interests
The authors declare no competing interests.
