## [Peer Review File · Nature Communications]

Reviewers' Comments:

Reviewer #1:

Remarks to the Author:

Literature Review

The following article addresses an important question involving the extent to which high performing women in math are less likely than high performing males in math to intend to pursue math as a career. While the analyses are interesting and the work an important addition to the existing literature, there are some issues that need to be addressed.

p. 1, 3rd paragraph: You seem to put a lot of emphasis on the fact that research demonstrating that girls have lower self-concept in math compared to boys (and more barriers to overcome) would lead everyone to conclude that women and girls need disproportionately higher grades than their male counterparts to have high enough self-concept to want to choose math as a career option. I'm not sure that I would agree that this is the only explanation to extrapolate from these findings or that researchers have settled on this as their key explanation. You can reason that this COULD be an explanation, but not necessarily that this is THE only explanation to infer from this finding. We know that when males and females are equally matched in math ability, boys tend to overestimate their skills and girls tend to underestimate them, but that doesn't necessarily mean that the girls with high ability have meaningfully higher enough self-concept than the girls with average ability that they will now pursue math as a result. Research has demonstrated that self-concept in math doesn't matter as much in determining career intentions as other factors. Math interest and career values (e.g. prefer working with people vs. objects; altruism) often are shown to matter more in determining math career intentions than math self-concept does. There is great appreciation in the literature for the complexity of the associations you are describing here, so your conclusion seems misleading and overly simplified. It is a possible explanation, not THE only conceivable explanation one can derive from these findings. Furthermore, there has been work that has illuminated that the highest achieving females in math are less likely than the highest achieving males to pursue STEM fields, mainly because they often are more likely to be high achieving in the language arts as well, drawing them into non-math intensive fields. Males, on the other hand, are more likely to have a stronger tilt toward high math skills (with lower relative language arts skills) restricting their choices to math-intensive fields. So this argument you are making that high achieving females in math choose STEM less often than high achieving males has already been demonstrated in the field, despite the fact that you present it as an entirely novel finding that hasn't been already examined and established. See the citations below:

Wang, M.-T., Degol, J. L., & Ye, F. (2017). Who Chooses STEM Careers? Using A Relative Cognitive Strength and Interest Model to Predict Careers in Science, Technology, Engineering, and Mathematics. *Journal of Youth and Adolescence*, 46, 1805-1820. doi:10.1007/s10964-016-0618-8
Wang, M. T., Eccles, J. S., & Kenny, S. (2013). Not lack of ability but more choice: Individual and gender differences in STEM career choice. *Psychological Science*, 24, 770-775. (which you cited in your paper)

Pages 6-7: when you reference the "gender gap in math performance" I found this terminology misleading. It makes me think you are describing an actual gap in math performance between the two genders (e.g., boys on average score 4 points higher than girls). In actuality, the gender gap in math performance you are describing is that there is a smaller proportion of higher performing females, relative to higher performing males intending to pursue math. I think to increase clarity, there could be a better description of this than "gender gap in math performance." I understand the need to be succinct, but I think this phrasing will confuse many readers who are accustomed to seeing it in the literature referencing a simple gender difference in average math performance in the population. This leads me to my second point, I assume when you calculated the girls-to-boys ratios in Table 2, you accounted for the difference in sample size for girls and boys. You may have mentioned it somewhere and I accidentally overlooked it, but I would appreciate an explanation of how these were calculated since you noted an uneven distribution of males and females in your sample.

p. 8; The last complete sentence on the page is awkwardly written. Revise to increase clarity.

p. 9; First complete sentence is also awkwardly written, "we find back" is an odd way to describe that your supplemental analyses aligned with your main analyses. Revise to increase clarity.

Table 1: Since these analyses are logistic regression models, are the numbers reported in the columns the coefficients or the odds ratios? I assume they are coefficients but having the odds ratios would be more useful for the reader. Also, are these models hierarchical, meaning that you kept adding more sets of covariates with each model, or does each model have a completely different set of covariates? If the latter is the case, what was the rationale for this decision? In particular, the fourth model where math interest is added appears to have the greatest proportion of variability in the outcome explained by the predictors, so what would be the rationale for then removing that covariate and running the other models without it?

p. 10; I don't necessarily agree with the conclusion that comparative advantage of math versus reading ability has no role in gender differences in pursuing math simply because the gap shrinks at the highest level of ability. I think what you are seeing could be a potential 3-way interaction effect (math ability x gender x math-reading balance) where math-reading balance may affect decisions to select math at the high ability end but not the low ability end. But since you didn't actually run regression analyses testing this 3-way interaction on intentions to choose math, it seems an overreach to say this factor does not matter. Your figures demonstrate two key findings: 1) When males have lower math ability they are more likely to have equally low ability in language arts, when females have lower math ability they are more likely to have higher relative language arts scores. 2) When males have higher math ability they are more likely to have much higher math scores relative to language arts scores, when females have higher math ability they are more likely to have equally high skills in language arts. I think this aligns with the previous research I cited for you in which cluster analyses of ability profiles show that a larger number of females have a more balanced set of skills at the high achieving end, and therefore, have more options to explore both math and non-math options, whereas the high achieving males restrict their choices more to the math options. So even just looking at your figures, the math-reading balance APPEARS to matter more at the high end of the math ability spectrum than at the low end, operating in expected ways established by previous literature. Until you run analyses and examine the potential interaction of math-reading balance x gender x math ability on intentions to pursue math, you cannot make such declarative statements. This goes beyond what your data show.

p. 11; The second sentence in the second paragraph is too long and confusing. Revise to increase clarity.

Reviewer #2:

Remarks to the Author:

The ms describes a series of analyses that examine how sex differences in intention to study in a math intensive field vary across levels (binned into 20 groups) of math performance. The results show that the difference in this intention increases with increases in math competence, which is interesting in and of itself; the effect is due to more rapid increases in boys' versus girls' intentions as competence increases. On the plus side, the analyses use several large (public) and rich data sets that show the same effect and include several control variables. After considering several alternatives, they conclude that gender stereotypes are driving the effect. However, the gender stereotype variable used here is not a good measure of the construct (below) and the authors fail to consider the utility of this construct in the broader literature (all of which was ignored), as well as other issues (below).

P. 2. The second assertion regarding the loss of talent may or may not be correct. This depends on what they are choosing for alternative careers. As has been documented in the Study of Mathematically Precocious Youth, women who are talented in mathematics have broader interests than their male peers and thus go into a wider variety of fields. Many women who are strong in math and science go into medicine instead of a math-intensive field. Is it a net social gain to encourage these talented women to move from medicine to computer science? In other words, this second point needs to be rewritten to better reflect these trade-offs or dropped; a similar assertion is made toward the end of this page.

Next paragraph and Discussion, regarding self-confidence and grades. Girls and women generally do better in school, including better grades, not just in mathematics and so this is not a domain-specific compensatory effect as implied here. Moreover, there is a large literature on the relation between factors such as self-efficacy or self-confidence as related to STEM engagement and any associated sex differences. There are often relations, but they are not particularly large; in any case, the citation here is 20 years old and doesn't capture the nuanced findings that have emerged since that time. One of the analyses in this ms appears to use the same data set as in the article below, but doesn't include interest variables and comes to a different conclusion (the conclusion below focused on the importance of interests, which are ignored in this ms):

Wang, M. T., Eccles, J. S., & Kenny, S. (2013). Not lack of ability but more choice: Individual and gender differences in choice of careers in science, technology, engineering, and mathematics. *Psychological Science*, 24, 770-775. doi:10.1177/0956797612458937

Similarly, here in the Introduction and later in the Discussion, there are now several meta-analyses that show that phenomena like stereotype threat on girls and women's math performance is very small and may not differ from no effect, once publication bias is considered. The proxy measure for stereotypes is not standard and not justified. In the PISA data, the correlation between students' perceptions of parental beliefs about the importance of math (one of the items used here) and parents' actual perceptions is only about .2 (across the countries with data on both) and thus is not likely to be a good indicator of external gender math stereotypes. The argument that higher SES parents are more stereotyped is based on a very old reference and otherwise not supported. Students perceive that their parents think math is more important for boys than girls in nations with higher GDP, but the within nation correlation between the PISA family wealth variable and what students believe about parent perceptions is weak and typically not-significant.

The finding that the sex difference in math-field intentions varies with level of math competence is interesting and relatively novel to my knowledge. However, girls with strong math scores also tend to have even better reading scores and their comparative advantage in reading may be influencing their career choices, not stereotypes or gender norms. This comparative advantage effect also applies to boys, whereby more boys than girls have higher math than reading scores, which will bias their career choices. The authors address this issue in the Discussion, which is important, but this should be in the analysis section and mentioned in the introduction.

For the analyses, it is not clear from the description in Table 1, but it appears that overall reading scores were used as a covariate which helps to address the issue, but I'm not sure that this fully captures the differences associated with women's comparative advantage in reading or the obverse of men's comparative advantage in math. This variable should be tried in lieu of overall reading.

Reviewer #3:

Remarks to the Author:

This paper employs data that are impressive in scope, spanning 61 countries and around 300,000 students, to investigate how teenagers' math ability and gender interact to predict their intentions to study math. The authors find that the gender gap in intentions to study math grows as math ability increases and is highest among the highest-performing students. Though the methods were impressive, I was left with many questions and confusions that prevent me from feeling like this paper is ready for publication.

First, I did not understand why the article needed to be framed around confidence because that construct did not seem to be measured. The authors link "girls' lower self-confidence" to "girls would need to perform better in order consider math education as a viable option" but those are not necessarily linked and there is not citation supporting that link. The story seems a lot simpler to focus on math ability and intentions to study math without bringing in the girls' lower confidence story. Moreover, this story is problematic because it reinforces the message that girls

are deficient in some way when in fact the greater deficiency may be boys' overconfidence about their abilities.

Second, I found these results not different enough from the Cimpian et al. Science paper showing that low-ability men tend to major in STEM to a greater extent than low-ability women. The authors distinguish their work from this previous work, but those seem more like incremental extensions (e.g., a bigger sample, investigation of stereotypes).

Third, I was confused about the difference between math self-concept and the data used to represent stereotypes about math aptitude. I could have used some clarification on how perceiving that "doing well in math is completely up to them" is distinct from math self-concept, especially since the authors argue that math self-concept does not explain the observed gender gap whereas the "doing well in math is completely up to them" item does.

Fourth, the "Fear of discrimination and male-dominated environments" section seemed to over-claim. For example, the authors demonstrate that high-math-ability girls do not report believing that women are discriminated against in science careers more than their low-math-ability peers. Could there be a difference between girls' evaluations of the reality of science careers vs. their own personal feelings about what it would be like to enter those fields? I.e., could higher math-ability girls be more hesitant to enter potentially discriminatory environments? The data seem to address perceptions of discrimination rather than fear of discrimination—certainly the two are linked, but they might not be identical. Moreover, science and math are two different subjects, asking about one might not generate the same answers as the other. Additionally, students were asked whether the prospect of being in an all-male or all-female environment was important to their choices, but it seems like all-male or all-female environments would be rare in the real world. What about majority-male and majority-female?

Fifth, there was inadequate discussion of culture in a project working with global data, except for a brief mention of developed countries having higher math performance and higher gender gaps in math intentions in the "Robustness checks" section. It is impressive to see the observed patterns operating worldwide! However, the paper could use more discussion on whether and why math performance, intentions, and especially the potential explanatory factors considered in the Discussion might be variable or constant across different cultures worldwide.

Finally, I wondered what "interiorizing" of the stereotype meant and whether that was necessary to the theory. Work by Steele and others has shown that negative stereotypes can affect aspirations and performance even if not internalized. Because there was no evidence that these stereotypes were endorsed, it may be more accurate for the authors to remove that claim.

Rebuttal letter

This letter includes

1. a general description of the changes made in the revised manuscript
2. our point-by-point responses to reviewers' comments (in bold characters), explaining how we addressed them in the revised manuscript.

[Redacted]

1. General description of the revision

We agree that our original manuscript suffered from an overall lack of clarity

In this revision,

- we make clear the advances of our findings (in the introduction): we clarify *[Redacted]* our approach is different from all other previous works
- we remove the framing around self-confidence (in the abstract and in the introduction): this framing is not necessary and is confusing. As suggested by Referee 3, we simply focus on girls' and boys' intentions to study math as a function of ability
- we focus on the main result and its consequences in terms of gender gap in math performance among students who are willing to do math in their career
- we discuss more clearly the role of comparative advantage in math (in particular, we now do it in the introduction)
- we reduce the importance of our analyses of stereotypes as a possible explanation for our main result, and present them in a shorter way and with greater caution.
- we clarify all other concerns raised by the referees

Concerning references, we suppressed 3 references that were related to discussions we removed and added 2 references related to comparative advantage: Wang et al. (2017) that was suggested by two referees and Stoet and Geary (2018).

Wang, M.-T., Degol, J. L., & Ye, F. (2017). Who Chooses STEM Careers? Using A Relative Cognitive Strength and Interest Model to Predict Careers in Science, Technology, Engineering, and Mathematics. *Journal of Youth and Adolescence*, 46, 1805-1820. doi:10.1007/s10964-016-0618-8

Stoet, G., Geary, D.C. (2018). The Gender-Equality Paradox in Science, Technology, Engineering, and Mathematics Education. *Psychological Science*, 29(4):095679761774171. DOI: [10.1177/0956797617741719](https://doi.org/10.1177/0956797617741719)

2. Point-by-point answers to Reviewers

Point-by-point answers to Reviewer #1 (Remarks to the Author):

Literature Review

The following article addresses an important question involving the extent to which high performing women in math are less likely than high performing males in math to intend to pursue math as a career. While the analyses are interesting and the work an important addition to the existing literature, there are some issues that need to be addressed.

We thank you for your encouraging remarks.

p. 1, 3rd paragraph: You seem to put a lot of emphasis on the fact that research demonstrating that girls have lower self-concept in math compared to boys (and more barriers to overcome) would lead everyone to conclude that women and girls need disproportionately higher grades than their male counterparts to have high enough self-concept to want to choose math as a career option. I'm not sure that I would agree that this is the only explanation to extrapolate from these findings or that researchers have settled on this as their key explanation. You can reason that this COULD be an explanation, but not necessarily that this is THE only explanation to infer from this finding. We know that when males and females are equally matched in math ability, boys tend to overestimate their skills and girls tend to underestimate them, but that doesn't necessarily mean that the girls with high ability have meaningfully higher enough self-concept than the girls with average ability that they will now pursue math as a result. Research has demonstrated that self-concept in math doesn't matter as much in determining career intentions as other factors. Math interest and career values (*e.g.*, prefer working with people vs. objects; altruism) often are shown to matter more in determining math career intentions than math self-concept does. There is great appreciation in the literature for the complexity of the associations you are describing here, so your conclusion seems misleading and overly simplified. It is a possible explanation, not THE only conceivable explanation one can derive from these findings.

We fully agree with you that we put too much emphasis in the abstract and in the introduction on self-confidence in order to introduce our analysis in the initial manuscript. It was just an attempt to provide an intuitive motivation but it was not necessary and misleading. We have therefore removed any reference to self-confidence in the abstract and in the introduction to frame our analysis.

It is more correct and simpler to focus on math ability and intentions to study math without referring to girls' lower self-confidence. This is what we do in the revised version. We have changed the abstract and the introduction accordingly.

In detail:

- In the abstract, suppression of

" Girls are known to be less self-confident in math-related fields, which is often believed to imply that they need better grades than boys to be willing to pursue studies in those fields."

- In the introduction, suppression of the paragraph:

"A common explanation put forward in the recent literature for the lower participation of girls in math-related fields is their lower self-confidence or self-concept in math (13). According to this explanation, girls would need to perform better in math in order to reach the same levels of confidence as boys and to be willing to consider math education

as a viable option. A similar reasoning can be made more generally regarding other mechanisms or barriers preventing girls to specialize in math-related fields: to be willing to overcome these barriers, such as stereotypes, social norms or (fear of) discrimination, girls would need a higher academic performance in math than boys. This simple theory of a trade-off between academic performance and other factors affecting educational decisions differentially for girls and boys rationalizes the idea that, since few girls are willing to choose math-related fields, it is likely that only the most able ones do so. Such a pattern of selection would lead in particular to a better average performance of girls compared to boys among students willing to enroll in math-related fields."

Furthermore, there has been work that has illuminated that the highest achieving females in math are less likely than the highest achieving males to pursue STEM fields, mainly because they often are more likely to be high achieving in the language arts as well, drawing them into non-math intensive fields. Males, on the other hand, are more likely to have a stronger tilt toward high math skills (with lower relative language arts skills) restricting their choices to math-intensive fields. So this argument you are making that high achieving females in math choose STEM less often than high achieving males has already been demonstrated in the field, despite the fact that you present it as an entirely novel finding that hasn't been already examined and established. See the citations below:

Wang, M.-T., Degol, J. L., & Ye, F. (2017). Who Chooses STEM Careers? Using A Relative Cognitive Strength and Interest Model to Predict Careers in Science, Technology, Engineering, and Mathematics. *Journal of Youth and Adolescence*, 46, 1805-1820. doi:10.1007/s10964-016-0618-8

Wang, M. T., Eccles, J. S., & Kenny, S. (2013). Not lack of ability but more choice: Individual and gender differences in STEM career choice. *Psychological Science*, 24, 770–775. (which you cited in your paper)

We agree that there exists previous work showing that females choose STEM fields at a much lower rate than males, and that it holds true at the top of the ability distribution, the highest achieving females in math being also less likely than the highest achieving males to pursue STEM fields. And comparative advantage is indeed one of the most promising reasons advanced (see below for specific answers to your comments about comparative advantage). We mentioned this literature in the original manuscript in the introduction (“Our work relates to the literature linking gender differences in academic performance to the gender gap in the pursuit of STEM or math-related studies (15-20)”). We agree that we should have added in the introduction the references you mention (even though we mentioned the second one later in the paper as you noticed) and we corrected our mistake in the revised version: we now cite both papers by Wang and coauthors in the introduction.

However, our argument is not the one you mention. The aim is not to show that highly able girls choose math fields less than highly able boys (maybe because of a lower tilt toward math as you mention¹), but that they do it less than low ability girls with respect

¹ More precisely, Wang et al. (2013) consider a sample of 671 US subjects interviewed by telephone in 1992 when they were in 12th grade and later in 2007 when they were 33 yo. All subjects were highly able in math in 12th grade. Wang et al. (2013) show that more males than females were in a STEM career at 33, and that the difference was reduced and not significant when controlling for the fact of being highly able in reading (but also for self confidence in math, in reading, for valuing working with things, with people, for math interest)

to low ability boys. Indeed, we analyze how the gender gap in choice of math fields evolves with the level of math ability and we show that it increases with ability and that the ratio of boys to girls choosing math increases with ability. This analysis has implications in terms of gender gaps in math performance as we show in the paper: the increase of the boys-to-girls ratio leads to a gender gap in math performance among students intending to do math that is higher than in the general population. The analysis along the ability distribution also permits to better understand the leaky pipeline, and more precisely to better identify where in the ability distribution it is most important, with possible policy implications for higher gender equality.

In the revised version, we tried to make it clearer that our approach is different from previous work since we focus on the behavior along the ability distribution:

- The first sentence of the abstract: " Using data on about 300,000 15-year-old students in 61 countries, we analyze boys' and girls' educational intentions along the ability distribution"

- In the introduction: " It is however different from that literature since we focus on gender differences in intentions to do math *along the math ability distribution* and are not interested in explaining the raw gender gap in math-related careers using various measures of ability."

-In the main text, several times, eg. at the end of the section "To wrap up, relative to boys, high-performing girls are less willing to do math than the less performing ones. This might induce a lower representation of high-performing women in math-fields, if educational intentions translate into actual choices."

Besides, we now devote a whole paragraph in the introduction to comparative advantage, with the references you mention.

Pages 6-7: when you reference the "gender gap in math performance" I found this terminology misleading. It makes me think you are describing an actual gap in math performance between the two genders (e.g., boys on average score 4 points higher than girls). In actuality, the gender gap in math performance you are describing is that there is a smaller proportion of higher performing females, relative to higher performing males intending to pursue math. I think to increase clarity, there could be a better description of this than "gender gap in math performance." I understand the need to be succinct, but I think this phrasing will confuse many readers who are accustomed to seeing it in the literature referencing a simple gender difference in average math performance in the population. This leads me to my second point, I assume when you calculated the girls-to-boys ratios in Table 2, you accounted for the difference in sample size for girls and boys. You may have mentioned it somewhere and I accidentally overlooked it, but I would appreciate an explanation of how these were calculated since you noted an uneven distribution of males and females in your sample.

We are in fact describing two patterns: first, a true performance gap between the two genders (difference between the scores of boys and the scores of girls) as the one you mention, and second, a gender imbalance among high performers.

The title of the section in the original manuscript that only mentioned the first may have been misleading.

Indeed, we show in Table 2 two results:

1. that the gender gap in math performance at the detriment of girls is higher among math choosers than in the whole population: the average score of boys is 0.19 higher than that

of girls among students intending to do math,² whereas it is 0.14 higher in the whole population, which corresponds to an "increase" of the gender gap of 40% at the detriment of girls. The girls who intend to do math were lower achievers than the boys who intend to do math.

2. that the representation of girls among high achievers is lower among math choosers than in the whole population (girls-to-boys ratio of 0.54 vs 0.74)

To avoid confusion, we now made it clear we consider the two different points:

- in the introduction: " These results imply that the gender gap in math performance is larger (in favor of boys) among students who intend to study math than among the general student population. They also imply a lower representation of high performing girls among students who intend to pursue math studies and careers, and both of these features may contribute to the survival of the stereotype that girls are less talented in math than boys."

- in the title of the section: " Implications for girls' lower math performance and lower representation among high performers"

- in the first sentence of the section: "The patterns highlighted above have implications both for the relative math performance of girls and for their representation among high performers in the subpopulation of students who intend to do math."

For your second point, there is no difference in representation of boys and girls in our initial sample. However, you are right that since 27% of the boys and 20% of the girls choose to do math, there is a difference of representation of boys and girls among math choosers. This is one of the reasons why we cannot interpret the raw decrease of the boys-to-girls ratio from 0.74 to 0.54 in Table 2 as the impact of the variation of girls' and boys' math choice with math ability only. The decrease of the boys-to-girls ratio is interesting in itself since it describes the representation of girls among high performers in the whole population and among math choosers, but as far as the impact of the differential selection of boys and girls is concerned, we write in the manuscript on p7 : *"Note that these patterns may be attributed to other factors than the variation of girls' and boys' math intentions with math ability pictured on Figure 1a. [...] the lower representation of women among high math performers among students intending to do math than in the general population is in part directly attributable to the fact that on average girls are less willing to study math than boys, implying that they are on average less represented in the former group."*

This is why in the following paragraph, we use a reweighting procedure, to take these 'other factors' into account and to guarantee comparisons that are neither driven by the proportion of boys and girls choosing math nor by the gender differences in math ability distribution. We show that we still obtain gender gaps in math performance that are higher among math choosers as well as girls-to-boys ratios that are lower in the top decile of math choosers than in the whole population, even if the effect is less important (see Table A6).

Finally, we also prove a theoretical result showing that if the girls-to-boys ratio of math choosers is decreasing with math ability (a condition that we show to be satisfied on our data), then the gender gap in math performance is higher among math choosers and the girls-to-boys ratio among top performers is lower among math choosers.

² gender gaps are expressed as a fraction of the standard deviation of math performance.

We tried to make these points clearer in the revised version: " The procedure guarantees that the comparisons are no longer driven either by the proportion of girls and boys choosing math or by the initial gender differences in the math ability distribution, and we still observe that the gender gap in math performance is higher (but less) among students intending to do math than in the general population and that the ratio of girls-to-boys decreases with math ability (but less) in the former group while it is by construction constant in the former one (Table A6). "

p. 8; The last complete sentence on the page is awkwardly written. Revise to increase clarity.
p. 9; First complete sentence is also awkwardly written, "we find back" is an odd way to describe that your supplemental analyses aligned with your main analyses. Revise to increase clarity.

We corrected the sentences and hope that they are clearer. Thank you

"For 10th graders, we also study intentions (when they are in 10th grade) to enroll in a scientific track the year after as well as their actual enrollment in such a track (a year after when they are in 11th grade). If we focus on actual enrollment in a scientific track, the results align with our main analyses (columns 2 and 3)."

Table 1: Since these analyses are logistic regression models, are the numbers reported in the columns the coefficients or the odds ratios? I assume they are coefficients but having the odds ratios would be more useful for the reader. Also, are these models hierarchical, meaning that you kept adding more sets of covariates with each model, or does each model have a completely different set of covariates? If the latter is the case, what was the rationale for this decision? In particular, the fourth model where math interest is added appears to have the greatest proportion of variability in the outcome explained by the predictors, so what would be the rationale for then removing that covariate and running the other models without it?

Table 1 provides the results of a linear probability model (i.e. a linear regression and not a logistic regression):

$$Math_intention_i = \alpha Girl_i + \beta Math_ability_i + \gamma(Math_ability_i * Girl_i) + \delta X_i + \varepsilon_i \quad (1)$$

The numbers in Table 1 indeed are the coefficients. The coefficient of interest is the number γ ("girls*(math performance)" in the table) representing the difference in slopes between the variation of girls' and boys' math choice with math ability. We essentially consider three different models, in addition to the model without control: one controlling for reading and science abilities, one controlling for self-confidence in math and one controlling for interest in math. None of them affects our results: the coefficient γ is unchanged in the last two models, and it is increased in the first model compared to the model without control.

We agree that since the variable is binary, we could have used a logistic model. We can easily provide the results if you deem it necessary. Nevertheless, we think the linear probability model is appropriate in our context and makes the interpretation of the results easier. Basically, the regression model provides the intercepts and slopes of the straight lines that best fit the data shown on our Figure 1. There is therefore a simple and direct link between what is shown on Figure 1 and the more formal regression analyses.

Note finally that in Table A7 col. 3 & 4, as well as in Table A2 col. 5-10, we replace the dummy for intending to do math, that is our main measure of math choice, by an index capturing more continuously students' intentions to do math, and the results remain valid.

p. 10; I don't necessarily agree with the conclusion that comparative advantage of math versus reading ability has no role in gender differences in pursuing math simply because the gap shrinks at the highest level of ability. I think what you are seeing could be a potential 3-way interaction effect (math ability x gender x math-reading balance) where math-reading balance may affect decisions to select math at the high ability end but not the low ability end. But since you didn't actually run regression analyses testing this 3-way interaction on intentions to choose math, it seems an overreach to say this factor does not matter. Your figures demonstrate two key findings: 1) When males have lower math ability they are more likely to have equally low ability in language arts, when females have lower math ability they are more likely to have higher relative language arts scores. 2) When males have higher math ability they are more likely to have much higher math scores relative to language arts scores, when females have higher math ability they are more likely to have equally high skills in language arts. I think this aligns with the previous research I cited for you in which cluster analyses of ability profiles show that a larger number of females have a more balanced set of skills at the high achieving end, and therefore, have more options to explore both math and non-math options, whereas the high achieving males restrict their choices more to the math options. So even just looking at your figures, the math-reading balance APPEARS to matter more at the high end of the math ability spectrum than at the low end, operating in expected ways established by previous literature. Until you run analyses and examine the potential interaction of math-reading balance x gender x math ability on intentions to pursue math, you cannot make such declarative statements. This goes beyond what your data show.

Actually, we must admit that we had the conviction at first that comparative advantage was related to the pattern that we identify. All the more so as two of us have recently worked on comparative advantage and choice of STEM fields³. We have therefore carefully verified if comparative advantage could play a role.

As you mention, we show in Figure 3 that the gap between boys' and girls' comparative advantage in math (i.e., the difference between the math and the reading score) shrinks at the highest level of ability. This shows that if we assume math intentions to depend linearly on comparative advantage for both girls and boys, then the latter variable cannot account for the behavior of the gender gap in math intentions along the ability distribution. This is confirmed by the results in Table 1 col. 2, where controls in the linear regression of the intentions to do math on gender, math ability and gender interacted with math ability include a third order polynomial in reading ability, a third order polynomial in science ability, reading ability interacted with gender, science ability interacted with gender, and interaction terms between math, science and reading ability (3 two-terms interactions and one three-term interaction). If we remove from the controls all terms involving science ability and only keep the terms involving math and reading, then the coefficient of gender interacted with math ability is equal to -0.029, to be compared to -0.021 in the setting without control (first column) and the impact of math ability on the gender gap is slightly increased. Also note that the results remain exactly the same if we replace the reading score by the difference between the math and reading score.

All this being said, your point is absolutely correct: the math-reading balance could play non-linearly and have a stronger effect for high-math-ability students. Furthermore, it is not entirely clear that our interaction term between math and reading in column 2

³ Breda, T., Napp, C. (2019). Girls' comparative advantage in reading can largely explain the gender gap in math-related fields. *Proceedings of the National Academy of Sciences*, **116**(31), 15435-15440.).

captures this appropriately. We have therefore tested your hypothesis directly and examined the effect of the interaction of math-reading balance x gender x math ability on intentions to pursue math. The model we adopt is saturated and includes 7 terms in total: 3 three variables above, their 3 two-way interactions, and their three-way interaction. We find that the math-reading balance has a strong effect on math intentions (as expected and already known in the literature), but that all other interaction terms including the math-reading balance are not statistically significant (despite our very large sample size). This means that the comparative advantage (i) does not play on intentions more for girls than for boys, and (ii) does not play on intentions more for high-math-ability (female or male) students than for low-math-ability (female or male) students. In addition, the coefficient of interest capturing how math ability plays differentially for girls on math intentions is barely affected by the addition of those controls. Therefore, your very valid hypothesis does not seem supported by the data. Finally, we have also examined if the comparative advantage could have a non-linear effect: to this aim we have replaced in our controls the difference between the math and reading performance by dummies capturing different percentiles of this difference. This allows us to directly capture non-linear effects.

In the revised version:

- we have put much more emphasis on comparative advantage. We now mention comparative advantage as a possible explanation in the introduction:

" Regarding mechanisms for our main result, high performing girls in math could shy away more from math fields than low performing ones because of differences in boys' and girls' comparative advantage in reading over math as a function of math ability. Or more generally, by the joint distribution of girls' and boys' performance in math, reading and science. Indeed, literature has shown that students choose a given field of study on the basis of their relative (rather than absolute) ability in that field with respect to other fields and that girls' comparative advantage in reading over math can account for their lower representation in STEM fields (11-14). We show that this is not the case and that comparative advantage or the trade-off between performance in math and in other fields cannot explain the differential pattern of selection of girls and boys into math-related fields along the ability distribution."

- we have developed the section about comparative advantage on p.10, including the answers to your question

" Comparative advantage. First, female students who are good at math are more likely than their male counterparts to be even better in reading and this comparative advantage of girls for reading over math can account for a large part of the raw gender gap in intentions to study math (11-14). Similarly, students' performance in reading and their comparative advantage for reading over math is likely to vary with their math ability, and it may do so differentially for girls and boys. As students' intentions to do math are in part driven by a trade-off between their math performance and their performance in other subjects, our main results could potentially be explained by the joint distribution of female and male students' performance in math, reading and science. We show that it is not the case. First, the gender gap (boys minus girls) in comparative advantage for math (defined as math ability minus reading ability) *decreases* with math ability (Figure 3a), implying that, if anything, this comparative advantage should lead to a *lower* gender gap in intentions to do math among high math-achievers than among low math-achievers. Second, when we include in equation (1) a rich set of control variables capturing reading and science abilities as well as their interaction with gender, the difference between girls and boys in the relationship linking math ability to math intentions is actually magnified

(Table 1, column 2). Third, even when we allow the effect of the comparative advantage for math to affect math intentions differentially according to both gender and math ability (by including in a regression model math ability, gender and comparative advantage as well as their 3 two-way interactions and their three-way interaction), the coefficient of interest capturing how math ability plays differentially for girls on math intentions is barely affected. Fourth, this coefficient is not affected either if we allow the comparative advantage for math to influence math intentions non-linearly. We do so by controlling either for a dummy variable indicating a positive comparative advantage or by quintiles of the comparative advantage and their interaction with students' gender."

p. 11; The second sentence in the second paragraph is too long and confusing. Revise to increase clarity.

Sorry but we could not identify the sentence, could you please indicate in more detail which one it is.

Point-by-point answers to Reviewer #2 (Remarks to the Author):

The ms describes a series of analyses that examine how sex differences in intention to study in a math intensive field vary across levels (binned into 20 groups) of math performance. The results show that the difference in this intention increases with increases in math competence, which is interesting in and of itself; the effect is due to more rapid increases in boys' versus girls' intentions as competence increases. On the plus side, the analyses use several large (public) and rich data sets that show the same effect and include several control variables. After considering several alternatives, they conclude that gender stereotypes are driving the effect. However, the gender stereotype variable used here is not a good measure of the construct (below) and the authors fail to consider the utility of this construct in the broader literature (all of which was ignored), as well as other issues (below).

We thank you for your positive remarks on the main result. We detail below our response to your criticism on gender stereotypes as well as on other issues.

P. 2. The second assertion regarding the loss of talent may or may not be correct. This depends on what they are choosing for alternative careers. As has been documented in the Study of Mathematically Precocious Youth, women who are talented in mathematics have broader interests than their male peers and thus go into a wider variety of fields. Many women who are strong in math and science go into medicine instead of a math-intensive field. Is it a net social gain to encourage these talented women to move from medicine to computer science? In other words, this second point needs to be rewritten to better reflect these trade-offs or dropped; a similar assertion is made toward the end of this page.

We agree with you that the wording might be misleading and we dropped all references to loss of talent in the revised manuscript:

- in the introduction, we replaced "Second, it represents a loss of talent that can reduce aggregate productivity (10-11) — if many talented girls shy away from math-intensive careers — leading to a shortage of workers with math-related skills at a time when the demand for such skills is increasing (12)." by " Second, it can lead to a shortage of workers with math-related skills at a time when the demand for such skills is increasing (10). "

- toward the end of the page, we replaced " The larger gender gaps in math performance among individuals who intend to study math mean that there is a *loss* of high-performing women" by "They also imply a lower representation of high performing girls among students who intend to pursue math studies and careers,"

Next paragraph and Discussion, regarding self-confidence and grades. Girls and women generally do better in school, including better grades, not just in mathematics and so this is not a domain-specific compensatory effect as implied here. Moreover, there is a large literature on the relation between factors such as self-efficacy or self-confidence as related to STEM engagement and any associated sex differences. There are often relations, but they are not particularly large; in any case, the citation here is 20 years old and doesn't capture the nuanced findings that have emerged since that time. One of the analyses in this ms appears to use the same data set as in the article below, but doesn't include interest variables and comes to a different conclusion (the conclusion below focused on the importance of interests, which are ignored in this ms):

Wang, M. T., Eccles, J. S., & Kenny, S. (2013). Not lack of ability but more choice: Individual

and gender differences in choice of careers in science, technology, engineering, and mathematics. *Psychological Science*, 24, 770-775. doi:10.1177/0956797612458937

We agree that we put too much emphasis in the abstract and in the introduction on self-confidence in order to introduce our analysis in the initial manuscript. It is not necessary and misleading, and we have removed it. It is more correct and simpler to focus on math ability and intentions to study math without referring to girls' lower self-confidence. This is what we do in the revised version.

In detail:

In the abstract, suppression of

" Girls are known to be less self-confident in math-related fields, which is often believed to imply that they need better grades than boys to be willing to pursue studies in those fields."

In the introduction, suppression of the paragraph:

"A common explanation put forward in the recent literature for the lower participation of girls in math-related fields is their lower self-confidence or self-concept in math (13). According to this explanation, girls would need to perform better in math in order to reach the same levels of confidence as boys and to be willing to consider math education as a viable option. A similar reasoning can be made more generally regarding other mechanisms or barriers preventing girls to specialize in math-related fields: to be willing to overcome these barriers, such as stereotypes, social norms or (fear of) discrimination, girls would need a higher academic performance in math than boys. This simple theory of a trade-off between academic performance and other factors affecting educational decisions differentially for girls and boys rationalizes the idea that, since few girls are willing to choose math-related fields, it is likely that only the most able ones do so. Such a pattern of selection would lead in particular to a better average performance of girls compared to boys among students willing to enroll in math-related fields."

Moreover, we now introduce the results about self-confidence in the discussion section with greater caution: "gender differences in math self-concept (i.e., how students perceive their math ability) have often been considered in the literature to explain for the gender gap in math enrollment (e.g., 13,28,3)."

Concerning interest, note that they were not ignored in the original manuscript. To make this point clearer, we have added a sentence in the Introduction: " Regarding mechanisms for our main result, [...]. We show that neither self-confidence in math, nor interest in math-related fields seem to account for the differential selection into math"

In the Discussion section, p.11, third paragraph, we explain that gender differences in interest do not seem to account for the pattern we identify:

" Similar results are obtained when we replace math self-concept by students' declared interest for math, which can be considered as a measure of their preferences. For this variable as well, the evolution of the gender gap along the ability distribution is opposite to that of the gender gap in choice of math-related fields (Figure A10), hence declared preferences do not seem to account for the differential selection into math (Table 1, column 4)."

Regarding the reference you provided, we wonder if Wang et al. (2017) is not more strongly related to the point you are making here. We should have cited this latter paper in the original manuscript and we now do it in the revised version, in the introduction and

when we discuss comparative advantage. Note, however, that neither Wang et al. (2013) nor Wang et al. (2017) uses the same data as the one we use in this paper.

Similarly, here in the Introduction and later in the Discussion, there are now several meta-analyses that show that phenomena like stereotype threat on girls and women's math performance is very small and may not differ from no effect, once publication bias is considered. The proxy measure for stereotypes is not standard and not justified. In the PISA data, the correlation between students' perceptions of parental beliefs about the importance of math (one of the items used here) and parents' actual perceptions is only about .2 (across the countries with data on both) and thus is not likely to be a good indicator of external gender math stereotypes. The argument that higher SES parents are more stereotyped is based on a very old reference and otherwise not supported. Students perceive that their parents think math is more important for boys than girls in nations with higher GDP, but the within nation correlation between the PISA family wealth variable and what students believe about parent perceptions is weak and typically not-significant.

Note first that what we study in the manuscript is not related to stereotype threat. We do not analyze how the activation of a stereotype related to math can impair the performance of girls. We analyze if the existence of an internalized stereotype by girls, i.e., the belief that math is not for them, that would be higher for high ability girls, could play a role in the fact that high ability girls shy away the most from math studies.

To measure stereotypes, we adapted the measure used in Breda et al. (2020) and proposed to measure gender stereotypes about math, i.e., beliefs that "math is not for girls", by the difference in agreement between girls and boys with two items in PISA survey: 'my parents think that math is important for my career' and 'doing well in math is completely up to me'. The idea is that systematic differences between boys and girls in these items are likely to capture stereotypes regarding math as appropriate for a career choice for a girl for the first item and stereotypes regarding innate aptitude for math, or talent for math for the second item. This is explained in the first paragraph of the Section on Gender stereotypes regarding math, p. 11.

Your remark about the first item is very relevant, however we know that there are discrepancies between explicit stereotypes and implicit ones. When parents are asked about considering and raising differently their sons or daughters, they usually answer negatively, although there are objective differences. The low correlation between parents' perceptions and the perceptions of their children can reflect these discrepancies. Nevertheless, and reassuringly, there is a gap between parents' beliefs regarding their sons and daughters, consistent with what we find among students (see the detailed analysis of Stoet et al., 2016). Additionally, the parents' beliefs are only available for 11 countries, making it practically difficult to use it for the present study that covers 61 countries.

We think that our measure captures stereotypes about math. We find interesting the fact that these stereotypes increase with math ability, and this feature is consistent with existing literature. We also find interesting the fact that when the stereotype measure is introduced as a control variable, the coefficient measuring the differential selection of boys and girls becomes not significant. Finally, we find the self-fulfilling mechanism appealing. But we feel that you are not fully convinced by our analysis on stereotypes. To answer your concerns, in the revised manuscript:

- we acknowledge that the measure of stereotype that we adopt is not standard: in the introduction we write " We must be cautious because we rely on a measure of gender stereotypes regarding math that is not standard"

- we are very cautious about our interpretation regarding stereotypes: for instance in the introduction : " We ultimately suggest that gender stereotypes regarding math could provide a plausible mechanism. We provide evidence suggesting that these gender stereotypes are more salient among high-ability students. We must be cautious because we rely on a measure of gender stereotypes regarding math that is not standard, but our results would provide an example of how gender stereotypes can persist over time even in the absence of objective gender differences"

- we have drastically reduced the importance of stereotypes as a possible explanation and focused on the main results. We do not mention anymore this explanation in the abstract. We have reduced the part on stereotypes in the introduction, in the discussion and in the conclusion.

The finding that the sex difference in math-field intentions varies with level of math competence is interesting and relatively novel to my knowledge. However, girls with strong math scores also tend to have even better reading scores and their comparative advantage in reading may be influencing their career choices, not stereotypes or gender norms. This comparative advantage effect also applies to boys, whereby more boys than girls have higher math than reading scores, which will bias their career choices. The authors address this issue in the Discussion, which is important, but this should be in the analysis section and mentioned in the introduction.

Thank you for your positive remark.

Concerning comparative advantage, we agree with you that it is important. We must admit that we had the conviction at first that comparative advantage was related to the pattern that we identify. All the more so as two of us have recently worked on comparative advantage and choice of STEM fields.

We gave more importance to comparative advantage in the revised version:

- we now devote a whole paragraph in the introduction to comparative advantage as a possible explanation: "Regarding mechanisms for our main result, high performing girls in math could shy away more from math fields than low performing ones because of differences in boys' and girls' comparative advantage (in reading over math) as a function of math ability. Or more generally, by the joint distribution of girls' and boys' performance in math, reading and science. Indeed, literature has shown that students choose a given field of study on the basis of their relative (rather than absolute) ability in that field with respect to other fields and that girls' comparative advantage in reading over math can account for their lower representation in STEM fields (11-14). We show that this is not the case and that comparative advantage or the trade-off between performance in math and in other fields cannot explain the differential pattern of selection of girls and boys into math-related fields along the ability distribution."

- we have extended the part on comparative advantage in the discussion section, and in particular have included results related to a question of Referee 1

" Comparative advantage. First, female students who are good at math are more likely than their male counterparts to be even better in reading and this comparative advantage of girls for reading over math can account for a large part of the raw gender gap in

intentions to study math (11-14). Similarly, students' performance in reading and their comparative advantage for reading over math is likely to vary with their math ability, and it may do so differentially for girls and boys. As students' intentions to do math are in part driven by a trade-off between their math performance and their performance in other subjects, our main results could potentially be explained by the joint distribution of female and male students' performance in math, reading and science. We show that it is not the case. First, the gender gap (boys minus girls) in comparative advantage for math (defined as math ability minus reading ability) *decreases* with math ability (Figure 3a), implying that, if anything, this comparative advantage should lead to a *lower* gender gap in intentions to do math among high math-achievers than among low math-achievers. Second, when we include in equation (1) a rich set of control variables capturing reading and science abilities as well as their interaction with gender, the difference between girls and boys in the relationship linking math ability to math intentions is actually magnified (Table 1, column 2). Third, even when we allow the effect of the comparative advantage for math to affect math intentions differentially according to both gender and math ability (by including in a regression model math ability, gender and comparative advantage as well as their 3 two-way interactions and their three-way interaction), the coefficient of interest capturing how math ability plays differentially for girls on math intentions is barely affected. Fourth, this coefficient is not affected either if we allow the comparative advantage for math to influence math intentions non-linearly. We do so by controlling either for a dummy variable indicating a positive comparative advantage or by quintiles of the comparative advantage and their interaction with students' gender."

- We prefer to leave the paragraph about comparative advantage in the discussion section because it does not seem to explain our main result.

For the analyses, it is not clear from the description in Table 1, but it appears that overall reading scores were used as a covariate which helps to address the issue, but I'm not sure that this fully captures the differences associated with women's comparative advantage in reading or the obverse of men's comparative advantage in math. This variable should be tried in lieu of overall reading.

Indeed, in Table 1 col. 2, controls in the linear regression of the intentions to do math on gender, math ability and gender interacted with math ability include a third order polynomial in reading ability, a third order polynomial in science ability, reading ability interacted with gender, science ability interacted with gender, and interaction terms between math, science and reading ability (3 two-terms interactions and one three-term interaction).

If we remove from the controls all terms involving science ability and only keep the terms involving math and reading, then the coefficient of gender interacted with math ability is equal to -0.029, to be compared to -0.021 in the setting without control (first column). Since we consider linear regressions, if we replace reading and reading interacted with gender by comparative advantage and comparative advantage interacted with gender, then the sum of the coefficients of gender interacted with math ability and of gender interacted with comparative advantage will be equal to -0.029. In both settings, the impact of math ability on the gender gap is slightly increased compared to the setting without control.

REFERENCES

- Breda, T., Jouini, E., Napp, C., Thebault, G. (2020). Gender stereotypes can explain the gender-equality paradox. *Proceedings of the National Academy of Sciences*, 117(49), 31063-31069
- Wang MT, Ye F, Degol JL. Who Chooses STEM Careers? Using A Relative Cognitive Strength and Interest Model to Predict Careers in Science, Technology, Engineering, and Mathematics. *J Youth Adolesc.* 2017 Aug;46(8):1805-1820.
- Stoet, G., Bailey, D. H., Moore, A. M., & Geary, D. C. (2016). Countries with higher levels of gender equality show larger national sex differences in mathematics anxiety and relatively lower parental mathematics valuation for girls. *PloS one*, 11(4), e0153857.

Point-by-point answers to Reviewer #3 (Remarks to the Author):

This paper employs data that are impressive in scope, spanning 61 countries and around 300,000 students, to investigate how teenagers' math ability and gender interact to predict their intentions to study math. The authors find that the gender gap in intentions to study math grows as math ability increases and is highest among the highest-performing students. Though the methods were impressive, I was left with many questions and confusions that prevent me from feeling like this paper is ready for publication.

We thank you for your encouraging remarks on the data and methods

First, I did not understand why the article needed to be framed around confidence because that construct did not seem to be measured. The authors link "girls' lower self-confidence" to "girls would need to perform better in order consider math education as a viable option" but those are not necessarily linked and there is not citation supporting that link. The story seems a lot simpler to focus on math ability and intentions to study math without bringing in the girls' lower confidence story. Moreover, this story is problematic because it reinforces the message that girls are deficient in some way when in fact the greater deficiency may be boys' overconfidence about their abilities.

We fully agree. The fact of framing in the abstract and in the introduction our approach around self-confidence is not necessary and generates confusion and problems. We also agree that it is simpler to focus on math ability and intentions to study math without referring to girls' lower self-confidence. This is what we do in the revised version. We thank you for your suggestion.

In detail :

- In the abstract, suppression of "Girls are known to be less self-confident in math-related fields, which is often believed to imply that they need better grades than boys to be willing to pursue studies in those fields."

The abstract now starts directly with: "Using data on about 300,000 15-year-old students in 61 countries, we analyze boys' and girls' educational intentions along the ability distribution. We show that relative to boys, high performing girls in math are those who shy away the most from math studies"

- In the introduction, suppression of the whole paragraph:

"A common explanation put forward in the recent literature for the lower participation of girls in math-related fields is their lower self-confidence or self-concept in math (13). According to this explanation, girls would need to perform better in math in order to reach the same levels of confidence as boys and to be willing to consider math education as a viable option. A similar reasoning can be made more generally regarding other mechanisms or barriers preventing girls to specialize in math-related fields: to be willing to overcome these barriers, such as stereotypes, social norms or (fear of) discrimination, girls would need a higher academic performance in math than boys. This simple theory of a trade-off between academic performance and other factors affecting educational decisions differentially for girls and boys rationalizes the idea that, since few girls are willing to choose math-related fields, it is likely that only the most able ones do so. Such a pattern of selection would lead in particular to a better average performance of girls compared to boys among students willing to enroll in math-related fields."

Second, I found these results not different enough from the Cimpian et al. Science paper showing that low-ability men tend to major in STEM to a greater extent than low-ability women. The authors distinguish their work from this previous work, but those seem more like incremental extensions (e.g., a bigger sample, investigation of stereotypes).

We disagree that our results are not very different from the Cimpian et al. Science paper. *[Redacted]*

We think this is important, and this is why we exceptionally initiated this appeal process. *[Redacted]* We did not want to engage in the criticism of a previous paper in the submitted manuscript and we acknowledge that our description of our contribution with respect to Cimpian et al. paper was not clear enough.

We make it clearer in the revised version of the manuscript:

In the Introduction: " On a sample of about 6,000 students attending a 4-year college in the US, (21) shows in particular that the lowest-achieving males major in PECS (physics, engineering or computer science) at surprisingly high rates relative to girls and that the ratio of boys to girls majoring in PECS is decreasing with math ability from about ten boys to one girl at the bottom to less than two boys to one girl at the top. These results are opposite to our own conclusions obtained here regarding the ratio of boys to girls intending to pursue math studies at 15 y.o. in 61 countries. This difference is likely explained by differences in the samples considered in the two studies. (21) restrict their sample to students attending a four-year college, a subsample that represents about 25% of all high schoolers, and that is not representative of the whole student population as girls, and especially low achieving ones in math, are overrepresented⁴. Indeed, as shown in (17), many girls go to college to study humanities ("when they are not STEM ready", i.e., when they have not taken enough high-school courses in science and math) whereas boys go to college mainly to pursue scientific studies. As argued above, focusing on a representative sample of high-schoolers seems more appropriate to understand the leaky pipeline (how and why girls disappear from math-intensive STEM careers). Beyond this core methodological difference that has important implications for the results, our paper also adopts a different perspective by analyzing both theoretically and empirically how the gender gap in math performance among math students depends on the way girls and boys select into math studies depending on their abilities. In contrast, (21) consider issues related to retention and attraction rates over time and these are issues we are not able to tackle with our data. "

Third, I was confused about the difference between math self-concept and the data used to represent stereotypes about math aptitude. I could have used some clarification on how perceiving that "doing well in math is completely up to them" is distinct from math self-concept,

⁴ In the subsample considered in (21), girls are overrepresented because they go more to college than boys (56% of girls in the subsample). This is especially the case at the bottom of the math ability distribution (59% of girls) and virtually none of these low performing girls in math major in math-related fields. This implies (somewhat artificially) a very high boys-to-girls ratio of low-achieving four-year college students majoring in math-related fields.

especially since the authors argue that math self-concept does not explain the observed gender gap whereas the “doing well in math is completely up to them” item does.

To measure stereotypes, we adapted the measure used in Breda et al. (2020) and proposed to measure gender stereotypes about math, i.e., beliefs that "math is not for girls", by the difference in agreement between girls and boys with two items in PISA survey: 'my parents think that math is important for my career' and 'doing well in math is completely up to me'. The idea is that systematic differences between boys and girls in these items are likely to capture stereotypes regarding math as appropriate for a career choice for a girl for the first item and stereotypes regarding innate aptitude for math, or talent for math for the second item. The first item we use about parents is part of the index of subjective norms in PISA. It reflects how the environment of the student views mathematics and gender differences in this item (controlling for math ability) likely reflect gender stereotypes in math in the environment of the students.

We agree with you that the second item shares similarities with self-confidence in math and that the differences are subtle. However, self-confidence and perceived control over success, although related, are two different constructs and the second item is about perceived control over success. We have added in the revised manuscript a note about the two items "Note that in PISA the first item belongs to the questions about 'subjective norms' while the second item belongs to the set of questions about 'perceived control over success in math'."

Self-confidence refers to one's beliefs in one's ability (here in math). In PISA it is measured through items such as: "I am not good at mathematics" or "I get good grades in mathematics" or "I have always believed that math is one of my best subjects".

The item we use refers to perceived control over success. In PISA, perceived control over success in math is measured through this item, as well as other items such as "If I put enough effort, I can succeed in math" or "I could perform well if I wanted".

There is a notion in the item we use of something that is beyond the students' control, something related to external factors that they cannot control. Gender differences in this item (controlling for math ability) possibly capture beliefs about the fact that boys are innately better or more talented at math, which is different from gender differences in self-confidence.

The difference in the evolution of the gender differences in self-concept and in this item as a function of ability can be seen as an illustration of the fact that they do refer to different notions/ constructs.

Fourth, the “Fear of discrimination and male-dominated environments” section seemed to overclaim. For example, the authors demonstrate that high-math-ability girls do not report believing that women are discriminated against in science careers more than their low-math-ability peers. Could there be a difference between girls’ evaluations of the reality of science careers vs. their own personal feelings about what it would be like to enter those fields? I.e., could higher math-ability girls be more hesitant to enter potentially discriminatory environments? The data seem to address perceptions of discrimination rather than fear of discrimination—certainly the two are linked, but they might not be identical. Moreover, science and math are two different subjects, asking about one might not generate the same answers as the other. Additionally, students were asked whether the prospect of being in an all-male or all-female environment was important to their choices, but it seems like all-male or all-female environments would be rare in the real world. What about majority-male and majority-female?

Thank you for those fair points. We agree with all of them and indeed we have not been careful enough in our initial writing of this section. As the section you refer to is an attempt to reject possible explanations for our main results, as these explanations we try to reject may not be the most obvious ones, and as the points you raise make our attempt to reject these explanations not very satisfactory, we chose to drop the corresponding section. This participates to our efforts to redraft the paper around its main contribution, which is the finding of a new pattern, that we observe worldwide, and the implications of this new pattern.

Fifth, there was inadequate discussion of culture in a project working with global data, except for a brief mention of developed countries having higher math performance and higher gender gaps in math intentions in the “Robustness checks” section. It is impressive to see the observed patterns operating worldwide! However, the paper could use more discussion on whether and why math performance, intentions, and especially the potential explanatory factors considered in the Discussion might be variable or constant across different cultures worldwide.

We added to comment our main results the following statement: “Further investigation shows no clear association between countries’ development (measured with GDP or HDI) or extent of equality (GINI) and the magnitude of the differential selection by gender according to ability. This confirms that the main pattern uncovered in this paper is widespread across the world and can be found in all types of cultural contexts (e.g. Greece, Korea, Russia, Brazil, Malaysia, New Zealand, Peru, etc.).”

Additionally, we clarified in the discussion section that our measure of stereotypes possibly explaining our main result varies both across students of different abilities and across countries, hence capturing differences across countries in culture and stereotypes: “This measure of stereotypes captures both variations occurring across countries and students of different ability within a country.”

The explanatory factors we consider are constant across different cultures worldwide.

Finally, I wondered what “interiorizing” of the stereotype meant and whether that was necessary to the theory. Work by Steele and others has shown that negative stereotypes can affect aspirations and performance even if not internalized. Because there was no evidence that these stereotypes were endorsed, it may be more accurate for the authors to remove that claim.

The notion of interiorizing stereotypes is different from their endorsement. The word "interiorizing" stereotypes referred in the original manuscript to the fact that girls have interiorized that math is not for them, in contrast to boys. It does not mean that they endorse gender stereotypes. These boys and girls, if asked, will probably not explicitly say that "girls cannot do well in math", that "math is more important for boys' career". However, boys more than girls implicitly believe that 'doing well in math is up to them' or that "their parents think that math is important for their career" (see above). Since the word "interiorizing" could be misleading, we chose to remove it. In the revised manuscript, we replaced it by other formulations to avoid confusion. We hope you find the new version clearer.

[Redacted]

Reviewers' Comments:

Reviewer #1:

Remarks to the Author:

I have read through the rebuttal of the authors and the critiques addressed. I think they did a fairly good job of addressing the majority of the concerns. Here, I simply mention areas where I think further emphasis or clarification need to be addressed.

Regarding the fact that you ran linear regression analyses instead of logistic regression on a categorically coded dependent variable, I think you should provide some justification along with citations on the appropriateness of this method. I understand that linear regression can be used on categorical dependent variables, but it is not always appropriate depending on the range of probabilities in your outcome. Furthermore, most people have been taught (myself included) that logistic regression is THE method for analyzing data on a categorical outcome variable. So some justification with appropriate citations should be provided.

I agree with the other reviewers that your stereotype measure is problematic and I do not agree with your argument that it is an appropriate or acceptable measure to use. Construct clarity is very important when considering research published at such a high level as this, and there is no basis in the literature that the two items you used are an acceptable proxy measure for stereotypes. Simply saying the measure is not standard is not acceptable in my viewpoint. The differences in responses for males and females may be due to internalized stereotypes but that doesn't mean that the items themselves are measuring internalized stereotypes. It is actually hard to pin down what they are measuring because there are only two items and they measure very different things. One seems to be measuring the adolescents' perception of the value (utility value) that their parents place on math, and the other seems to be measuring either the adolescents' own self-concept in math, locus of control in math, or implicit mindsets in math. That is why you need measures with a larger range of items, so that you can better pinpoint what they are actually measuring. I would recommend removal of the stereotype variable from the analyses altogether and from any mention in the manuscript, save that you do not have a good measure of it, and therefore, cannot adequately examine this as a contributing factor.

In response to the rebuttal that your paper is very different from the Cimpian et al. paper published in Science, I agree with that point, but I do not necessarily agree that your findings are more correct than theirs. I do appreciate that you re-examined the data from the Cimpian study looking at the ratios of boys to girls in PECS from the total sample of high schoolers and demonstrated that the proportion of boys to girls in the lower performing math distribution drops significantly. I think that is an important point to make, and in that sense, the Cimpian findings might be unintentionally misleading. However, I think there is a key distinction that may also make your findings unintentionally misleading as well: the fact that your outcome measure included items that asked about math in a general way and did not specify the types of majors or careers that are considered to be "math-oriented." Throughout your manuscript you repeatedly reference STEM so I assume that when students indicated that they intend to study math you made the assumption that they are specifically referring to studying math in math-intensive STEM fields, but many non-STEM fields have majors that include the study of math at their core, and it is well understood in the literature that women are far more likely to pursue those fields than the math-intensive fields in STEM. For example, many of the 15-year-old girls in the PISA sample who are lower performing in math but stated that they are planning to study math, may actually be intending to become math teachers. We know that slightly higher percentages of math education majors at the undergraduate level are women, and it's possible that the adolescents in the PISA study that are planning to become math teachers may indicate their high intentions to study math. However, education falls outside of the realm of STEM and into the social sciences, which are overly populated by women relative to men. The same could be said for accountants. I've seen percentages that women are highly represented among accounting professions, and these professions also utilize a lot of math but fall outside of the realm of STEM. So while I appreciate that your re-examination of the Cimpian study shows they may have inadvertently overestimated the ratio of boys-to-girls among low math achievers majoring in PECS, your choice of outcome may have also inadvertently overestimated the ratio of girls-to-boys among low math achievers pursuing math IF your view of a math field is a STEM one (ex: engineering). The fact that you

overestimated may not necessarily be the case as the statistics I mentioned above about higher proportions of math education majors being female does not address gender proportions among the general population along an ability distribution. But it could explain why you found a larger ratio of girls-to-boys at the lower achieving end (where maybe girls are more likely to pursue non-STEM math fields) than the higher achieving end (where maybe girls are more likely to pursue the STEM math fields). It's a suggestion, but I think it needs to be addressed. But still, kudos on taking into account the full population in your analyses.

Reviewer #2:

Remarks to the Author:

The ms is a revision of a prior submission and the authors have addressed some, but not all the initial issues. The strengths include the use of several large data sets that show that the intention to pursue a math-intensive major in college (which they show is correlated with later enrollment) increases with increases in math competence, but the increase is stronger for boys than girls. This part is interesting in and of itself and appears robust. They then show that the differences in slope comparing boys and girls is not substantially affected by PISA items that tap math interest and math self-confidence.

The gap however disappears when they include a stereotype measure, which is the difference between a measure of how much math is important to them vs. important for their parents (i.e., their beliefs about whether their parents think math is important for them). The latter increases with increases in math performance and parental SES, leading to the conclusion that math stereotypes are stronger among higher achievement girls and this in turn suppressed their intentions to pursue math-intensive college course work.

By this logic, shouldn't higher achieving girls' math self-concept also be suppressed by their parents' attitudes? Rather, the authors find that girls' math self-concept increases with increases in math competence and does so more rapidly than that of boys.

This is a critical issue, given the focus on stereotypes as a potential causal factor dissuading high-achieving girls from going into math-intensive careers. Moreover, the favored regression model showing that the stereotype measure eliminates the core interaction (i.e., the above-mentioned slope difference across boys and girls) explains less than 3% of the variation in the intention to pursue a math-intensive field of study. There are clearly other factors that influence these intentions that are not captured or discussed here. One obvious one is the well documented sex difference in career aspirations, which is not the same as interest in math. Girls who excel in math and science often opt for careers in the life sciences, not more math-intensive careers.

In short, the issues regarding the stereotype measure remain, even though the authors are now more cautious in their interpretation, and even if the measure was a robust assessment of the influence of stereotypes, it adds little to the overall prediction of intentions to study math.

Reviewer #3:

Remarks to the Author:

I was a reviewer on the previous version of this paper and see that it has improved a lot in clarity by removing distracting arguments and more carefully describing the uniqueness of the work. My sense is that the authors are demonstrating something interesting but I'm still wondering whether this effect is framed in a way that best fits with the existing literature.

The relationship they uncovered between math performance and intentions seems to be part of another story where gender gaps may be wider among communities that are more privileged. Others who have shown that relationship include Maria Charles and Sean Reardon. Because math performance and SES are conflated, I wondered whether the main story here is really about gender gaps and SES (either on an individual level, country level, or both). Relatedly, I wondered about the role of individualism in these countries in shaping this gap based on Maria Charles' work

suggesting that greater individualism may predict greater STEM gender gaps. SES and individualism are not controlled in the analyses, but I did see SES discussed when thinking about mechanism. My thinking is rather than having SES be a mechanism, SES may be the independent variable that then shapes gender stereotypes which then shapes aspirations. These are all difficult to prove with the current cross-sectional data, but I would have liked more alternate models considered that include SES and also individualism (on the country-level).

One smaller issue is that I feel like the title is misleading because it does not specify that the comparison is relative to boys. Readers may assume that high-performing girls shy away from math more than low-performing girls, which would not be an accurate interpretation.

From eye-balling the graph, there also seems to be a small deviation at the end where the very top performers have a smaller gender gaps than those who are strong but not the very top performers. I wondered whether the authors speculate that their preferred mechanism – gender stereotypes – may play less of a role for those who are exceptional math performers.

Answers to Reviewer 1

I have read through the rebuttal of the authors and the critiques addressed. I think they did a fairly good job of addressing the majority of the concerns. Here, I simply mention areas where I think further emphasis or clarification need to be addressed.

Thank you very much for your encouraging remarks

Regarding the fact that you ran linear regression analyses instead of logistic regression on a categorically coded dependent variable, I think you should provide some justification along with citations on the appropriateness of this method. I understand that linear regression can be used on categorical dependent variables, but it is not always appropriate depending on the range of probabilities in your outcome. Furthermore, most people have been taught (myself included) that logistic regression is THE method for analyzing data on a categorical outcome variable. So some justification with appropriate citations should be provided.

We agree with you that logistic regressions are commonly used for a categorical dependent variable. We instead use a linear probability model (see reference below). As you rightly underline, a drawback of the linear probability models is that predicted probabilities can go outside the range [0,1]. Nevertheless, our understanding is that more recently, these models have been considered as appropriate as others, such as logit or probit, which also require some parametric assumptions that may be violated, in order to keep predicted probabilities between 0 and 1.

Fundamentally, all those models make parametric assumptions on the relationship between the explanatory variable (math ability in our case) and the dependent variable (a dummy for strongly intending to do math in our case). One way to decide which model can be the most appropriate is to examine non-parametrically how the function that relates the outcome to the dependent variable looks like. This is what we do on Figure 1, where we plot the “full function” that relates intentions to do math to fractiles of standardized math ability for girls and boys. We have similarly plotted intentions to do math directly as a function of standardized math ability (to do so, we average intentions to do math by bins of equal size of math ability). The results is shown below for girls and boys together:

Results for girls and boys taken separately are similar and show that the relationship that links math ability to intentions to study math is close to linear. This provides some justification to use the linear probability model as a baseline.

One advantage of the linear probability model is also that the regression coefficients can directly be interpreted as marginal effects and are therefore easier to read. In our case, we can also more directly link the regression results to the Figures of the paper. For example, our first linear regressions fit the slopes of the relationship between math performance and intentions to do math for girls and boys, something which is close to what we show on the various Figures. Hence, using linear probability models increases the readability of the paper.

Based on the discussion above, we have kept the linear specifications as a baseline, but have replicated our main result in the Appendix using a logistic model (Table A3), in order to show that it is not driven by model specification. We also added the following reference in the paper to justify the use of the linear probability model:

Wooldridge, Jeffrey M. (2013). "A Binary Dependent Variable: The Linear Probability Model". *Introductory Econometrics: A Modern Approach* (5th international ed.). Mason, OH: South-Western. pp. 238–243.

I agree with the other reviewers that your stereotype measure is problematic and I do not agree with your argument that it is an appropriate or acceptable measure to use. Construct clarity is very important when considering research published at such a high level as this, and there is no basis in the literature that the two items you used are an acceptable proxy measure for stereotypes. Simply saying the measure is not standard is not acceptable in my viewpoint. The differences in responses for males and females may be due to internalized stereotypes but that doesn't mean that the items themselves are measuring internalized stereotypes. It is actually

hard to pin down what they are measuring because there are only two items and they measure very different things. One seems to be measuring the adolescents' perception of the value (utility value) that their parents place on math, and the other seems to be measuring either the adolescents' own self-concept in math, locus of control in math, or implicit mindsets in math. That is why you need measures with a larger range of items, so that you can better pinpoint what they are actually measuring. I would recommend removal of the stereotype variable from the analyses altogether and from any mention in the manuscript, save that you do not have a good measure of it, and therefore, cannot adequately examine this as a contributing factor.

We fully understand your concerns, as well as those of the other referees and of the editor, regarding the stereotype measure. We followed your suggestion and removed the stereotype measure from our analysis and from any mention in the manuscript.

In response to the rebuttal that your paper is very different from the Cimpian et al. paper published in Science, I agree with that point, but I do not necessarily agree that your findings are more correct than theirs. I do appreciate that you re-examined the data from the Cimpian study looking at the ratios of boys to girls in PECS from the total sample of high schoolers and demonstrated that the proportion of boys to girls in the lower performing math distribution drops significantly. I think that is an important point to make, and in that sense, the Cimpian findings might be unintentionally misleading. However, I think there is a key distinction that may also make your findings unintentionally misleading as well: the fact that your outcome measure included items that asked about math in a general way and did not specify the types of majors or careers that are considered to be "math-oriented."

Throughout your manuscript you repeatedly reference STEM so I assume that when students indicated that they intend to study math you made the assumption that they are specifically referring to studying math in math intensive STEM fields, but many non-STEM fields have majors that include the study of math at their core, and it is well understood in the literature that women are far more likely to pursue those fields than the math intensive fields in STEM. For example, many of the 15-year-old girls in the PISA sample who are lower performing in math but stated that they are planning to study math, may actually be intending to become math teachers. We know that slightly higher percentages of math education majors at the undergraduate level are women, and it's possible that the adolescents in the PISA study that are planning to become math teachers may indicate their high intentions to study math. However, education falls outside of the realm of STEM and into the social sciences, which are overly populated by women relative to men. The same could be said for accountants. I've seen percentages that women are highly represented among accounting professions, and these professions also utilize a lot of math but fall outside of the realm of STEM. So while I appreciate that your reexamination of the Cimpian study shows they may have inadvertently overestimated the ratio of boys-to-girls among low math achievers majoring in PECS, your choice of outcome may have also inadvertently overestimated the ratio of girls-to-boys among low math achievers pursuing math IF your view of a math field is a STEM one (ex: engineering). The fact that you overestimated may not necessarily be the case as the statistics I mentioned above about higher proportions of math education majors being female does not address gender proportions among the general population along an ability distribution. But it could explain why you found a larger ratio of girls-to-boys at the lower achieving end (where maybe girls are more likely to pursue non-STEM math fields) than the higher achieving end (where maybe girls are more likely to pursue the STEM math fields). It's a suggestion, but I think it needs to be addressed. But still, kudos on taking into account the full population in your analyses.

Thank you for your comments about our reexamination of Cimpian et al.'s data. We fully agree with you regarding the fact that our focus on intentions to study math or pursue a path related career is different from PECS or STEM enrolment. Based on your comments, we have tried to do a better job to explain what our measure captures. To do so, we have first included your comment in our revised version when introducing our measure of intentions to do math (p. 5):

“Our main variable captures students' *intentions* to pursue math-related studies rather than actual enrollment. This approach facilitates world-level analyses. However, it is not entirely clear that intentions do indeed materialize into actual choice, nor that the corresponding choices lead to careers that fall in the realm of PECS or STEM. Students that intend to pursue math-related studies may for example become math teachers or accountants, two jobs that would not be counted within STEM careers.”

More importantly, we motivate our approach by the fact that it is well suited to study the leaky pipeline. First, the underrepresentation of females is largest in the most math-intensive fields of study (math, computer science, physical science, geosciences, engineering and economics). Hence focusing on math makes sense. Second, a large share of the enrollment in math classes at university is explained by courses taken in high school (Card and Payne, 2021). Hence, focusing on educational intentions at 15 y.o., when the first important educational decisions are made, is adapted to capture the roots of the leaky pipeline. These points are detailed in our introduction on p. 3:

“Note that our main variable of interest is aimed at capturing students' intended behaviors rather than actual enrollment. This variable has the advantage of being defined similarly in all countries, hence allowing for world-level analyses. Such analyses would be hard to perform with enrollment data that would need to be harmonized across several different educational systems and could be more directly affected by demand-side effects (e.g., number of seats available in math-related fields, or discrimination against girls or boys applying in some fields). Importantly, by considering other data sources on actual students' enrollment as well as previous works in specific countries (France, USA, Canada, Australia), we show that our results obtained on PISA2012 data on intentions to pursue math studies and careers likely extend to students' actual enrollment. Our choice of focusing on high-school students is also motivated by two points. First, it allows us to obtain results valid for virtually all students. Second, the choices of university degree programs are shaped by earlier subject choices in high school (15-16) and the leaky pipeline starts at the end of high school (17-18). As clearly explained by (17), a large share of the gender gap in STEM enrollment at university or in college is explained by STEM courses taken in high school. This implies that selection into math studies or intentions to take math courses *during high school* are likely to be a good predictor of the gender gap in math-related studies and careers later on.”

To clarify further that we do not focus on STEM, we have also been careful in our revision about mentioning STEM only when we cite other studies or results that concern explicitly STEM in general.

Finally, regarding more directly the comparison with Cimpian et al., we have added a footnote to explain why we do not think that the outcome variable considered explain the difference between our results with theirs. See footnote 3 on p. 4:

“Another obvious reason for the different result reached by (21) is that they focus on actual enrollment in PECS majors while we focus on intentions to do math. While we cannot fully discard it, we think this explanation is less likely, as we show that our results do extend to actual enrolment in a number of countries, including the U.S., when (course) enrolment is measured among a representative sample of high schoolers (see results below).”

In the same spirit, we also explain on p. 5 that our results regarding intentions extend to (i) actual enrolment in a number of countries and (ii) math-related (and STEM in this specific case) careers. See p. 5:

“To discuss these points, we first retrieved country-level data collected by the UNESCO on the proportion of graduate students who graduate in a STEM field to show that these proportions are broadly comparable in terms of magnitude to the country-level averages of our binary measure of math intentions (Table A1). More importantly, in what follows, we exploit a dataset containing information on both students’ intentions and choices in France to show that in this country in particular, intentions and enrollment are highly correlated, and that our conclusions hold when using actual enrollment instead of intentions. We also exploit data from the High School Longitudinal Study (HSL:09) in the United States as well as previous work relying on longitudinal datasets in Canada (17) or Australia (25) to suggest that our conclusions extend to actual enrollment. We finally exploit data on adults’ math skills and occupations to further suggest that our conclusions may also extend to math-related or STEM occupations.”

The results mentioned above are then described on p. 8-10. We made a few additional edits there to adjust the text to your comment. In particular, we highlight that the selection pattern documented in the paper is also observed in France for enrolment in selective (and highly prestigious) math-related higher education programs (see Table A9, column 7). This shows for this specific country that it is not entirely driven by the low-demanding or less prestigious math-related studies or careers. See footnote 8 on p. 9:

“This last result shows that our conclusions for France are found when we focus on the most demanding and prestigious math-related higher-education programs. Therefore, they are not driven exclusively by girls being over-represented among low-achieving math students going to educational programs that are low-demanding in math (and may potentially fall outside the realm of STEM or PECS studies).”

We think that altogether the existing evidence we provide or discuss regarding actual enrolment in the U.S., Canada, Australia and France supports the idea that our findings are likely to extend to enrolment in math-related high-school courses or higher education programs. We hope you will find these revisions satisfactory. In particular, we tried to have the good balance between benchmarking against Cimpian et al. paper and highlighting the main independent contributions of the present paper, which are (as we try to state clearly in the paper) to focus on a large number of countries, on representative samples of high-schoolers, and at the beginning of the leaky pipeline, which is known to matter a lot for female underrepresentation in STEM or PECS.

Thank you finally for your positive comment on the fact that we are able to take into account the full population in our analyses.

Answers to Reviewer 2

The ms is a revision of a prior submission and the authors have addressed some, but not all the initial issues. The strengths include the use of several large data sets that show that the intention to pursue a math-intensive major in college (which they show is correlated with later enrollment) increases with increases in math competence, but the increase is stronger for boys than girls. This part is interesting in and of itself and appears robust. They then show that the differences in slope comparing boys and girls is not substantially affected by PISA items that tap math interest and math self-confidence.

The gap however disappears when they include a stereotype measure, which is the difference between a measure of how much math is important to them vs. important for their parents (i.e., their beliefs about whether their parents think math is important for them). The latter increases with increases in math performance and parental SES, leading to the conclusion that math stereotypes are stronger among higher achievement girls and this in turn suppressed their intentions to pursue math-intensive college course work. By this logic, shouldn't higher achieving girls' math self-concept also be suppressed by their parents' attitudes? Rather, the authors find that girls' math self-concept increases with increases in math competence and does so more rapidly than that of boys.

This is a critical issue, given the focus on stereotypes as a potential causal factor dissuading high-achieving girls from going into math-intensive careers. Moreover, the favored regression model showing that the stereotype measure eliminates the core interaction (i.e., the above-mentioned slope difference across boys and girls) explains less than 3% of the variation in the intention to pursue a math-intensive field of study. There are clearly other factors that influence these intentions that are not captured or discussed here. One obvious one is the well documented sex difference in career aspirations, which is not the same as interest in math. Girls who excel in math and science often opt for careers in the life sciences, not more math-intensive careers.

In short, the issues regarding the stereotype measure remain, even though the authors are now more cautious in their interpretation, and even if the measure was a robust assessment of the influence of stereotypes, it adds little to the overall prediction of intentions to study math.

We understand your concerns regarding our analysis of the influence of stereotypes. They are also shared by other reviewers. Based on these concerns, the Editor asked us to remove from the paper our analyses of the role of stereotypes. The revised paper therefore focuses mostly on the part that you found interesting in itself and robust.

The role of stereotypes is still discussed in the mechanism section as one possible explanation for our findings, but we do not try anymore to provide a measure of these stereotypes and see if it can explain our main results.

Additionally, we acknowledge that gender stereotypes regarding math are multi-dimensional and may affect girls' and boys' self-concept or interest for math. We think this is what you have in mind when you write "By this logic, shouldn't higher achieving girls' math self-concept also be suppressed by their parents' attitudes?". We have clarified that the mechanisms we discuss are not mutually exclusive: stereotypes may influence math self-confidence, but they eventually influence decisions in several ways, and we do not need to observe the same patterns for self-confidence and intentions to study math to argue that stereotypes can explain the second pattern.

Finally, following your comments and those of other reviewers, we have also tried to clarify the discussion by focusing it specifically on gender stereotypes regarding excellence in math, that is the belief that excellence in math requires to be a genius or to have innate talent, something that is less (or not) likely for girls than boys. This specific stereotype has been described by the literature and we think it is more directly related to the fact that top female math performers are those that shy away the most from math relative to their male counterparts.

We hope you will find this revised version satisfactory.

Answers to Reviewer 3

I was a reviewer on the previous version of this paper and see that it has improved a lot in clarity by removing distracting arguments and more carefully describing the uniqueness of the work. My sense is that the authors are demonstrating something interesting but I'm still wondering whether this effect is framed in a way that best fits with the existing literature.

Thank you for your encouraging remarks

The relationship they uncovered between math performance and intentions seems to be part of another story where gender gaps may be wider among communities that are more privileged. Others who have shown that relationship include Maria Charles and Sean Reardon. Because math performance and SES are conflated, I wondered whether the main story here is really about gender gaps and SES (either on an individual level, country level, or both). Relatedly, I wondered about the role of individualism in these countries in shaping this gap based on Maria Charles' work suggesting that greater individualism may predict greater STEM gender gaps. SES and individualism are not controlled in the analyses, but I did see SES discussed when thinking about mechanism. My thinking is rather than having SES be a mechanism, SES may be the independent variable that then shapes gender stereotypes which then shapes aspirations. These are all difficult to prove with the current cross-sectional data, but I would have liked more alternate models considered that include SES and also individualism (on the country-level).

Thank you for these very useful comments. Following the Editor's request and the comments made by the other reviewers, we kept our initial framing and focus the paper on the differential selection of girls and boys in math-related studies along the math ability distribution. We also think that the empirical pattern we show is striking and widespread across countries, so it is worth describing in its own, and starting from that point, even if the pattern is partly explained by other factors such as SES. The main novelty and contribution of our paper is to document thoroughly in a large sample of countries how the gender gap in math intentions varies with ability. As you rightly point, SES has already been studied, and focusing on SES directly would not make such a contribution. Our approach also allows us to make the point that is that the selection of girls and boys along the ability distribution has implications for the gender gap in math performance among students that intend to do math, something that in its own can reinforce stereotypes regarding math performance, as we argue in the paper. Finally, controlling for SES in our empirical analyses only reduces by half the differential selection of girls and boys into math studies along the ability distribution. This means that there is more than just differences across SES groups in our findings, providing an additional reason for focusing on the role of ability.

All this being said, we think that your argument that “SES may be the independent variable that then shapes gender stereotypes which then shapes aspirations” deserves a clear discussion. We are well aware of the pioneering work of Maria Charles and Sean Reardon as it inspired us a lot in former work. In the revised manuscript, we have tried to link more directly our findings to this work when we discuss the role of SES and stereotypes. There is now a dedicated subsection that is as follows on p. 12:

“Socioeconomic background, educational environment and transmission of gender norms. The gender gap in intentions to study math is actually larger among high socioeconomic status (SES) students. Controlling for socioeconomic background and its differential effect on girls and boys in equation (1) actually lowers γ by about 50% (Table 1, column 5). This shows that our main results partly capture the fact that high math-performers tend to be predominantly from a high SES background. Such a finding is also in line with a possible influence of gender stereotypes. Indeed, it has been shown that high-income parents spend more time and money on their children, and to invest in more stereotypical activities (38). This may lead them to transmit earlier and to a larger extent gender norms (39), including those related to math careers (40). At a more macro level, gender gaps in math education are known to be larger in socioeconomically advantaged school districts (39) and in richer countries (41-42). Stereotypes have been advocated to be a possible explanation for these results (27, 39, 43) and could similarly explain our main findings. In particular, the type of educational environment that is likely to surround high math performers within a country, or prevail among more privileged school districts or more developed countries in general typically implies more difficult curricula, higher performance standards, and greater competition, all of which have been shown to heighten gender-essentialist ideas about math and science (44).”

Finally, we already mentioned in the manuscript on p. 6 the following:

“Further investigation shows no clear association between countries’ development (measured with GDP or HDI) or extent of equality (GINI) and the magnitude of the differential selection by gender according to ability. This confirms that the main pattern uncovered in this paper is widespread across the world and can be found in all types of cultural contexts (e.g. Greece, Korea, Russia, Brazil, Malaysia, New Zealand, Peru, etc.).”

We tried some additional cross-country analyses to see if the differential selection by gender according to ability was larger in more individualist countries, but could not identify any clear patterns. We however think that there is no reason why we should find a clear pattern. Indeed, there is a clear parallel between what we observe among students of different abilities or SES, and what has been observed at a more macro level among countries of different wealth, levels of equality or extent of individualism. The general idea is, as you rightly explain, that “gender gaps may be wider among communities that are more privileged.” However, the differences across ability or SES groups do not necessarily have to be magnified in some countries. Rather, they may explain the cross-country differences: in richer countries, students’ math ability is actually better, which provide a direct link between our more micro findings and the former cross-country literature. We have tried to mention this while keeping the text relatively short.

One smaller issue is that I feel like the title is misleading because it does not specify that the comparison is relative to boys. Readers may assume that high-performing girls shy away from math more than low-performing girls, which would not be an accurate interpretation.

We fully agree with you and changed the title.

From eye-balling the graph, there also seems to be a small deviation at the end where the very top performers have a smaller gender gaps than those who are strong but not the very top

performers. I wondered whether the authors speculate that their preferred mechanism – gender stereotypes – may play less of a role for those who are exceptional math performers.

What happens at the very top on the graph depends to some extent on the number of underlying percentiles we use and is not statistically meaningful. For this reason, we prefer not insisting too much on this. To nevertheless answer your question: we do not think that our preferred mechanism (gender stereotypes) may play less of a role at the top. This is in part because we argue that these stereotypes are likely to impact more girls performing well in math. As we are already prudent in this revision regarding our proposed explanations for the general pattern we document, we prefer not making additional hypothesis regarding what may happen at the very top.

Reviewers' Comments:

Reviewer #1:

Remarks to the Author:

I have reviewed the manuscript. Overall I think the authors did a great job addressing the concerns raised by the editor and the reviewers. My main two notes for revision are as follows:

1) Regarding discussion of stereotypes as a potential mechanism that explains increasing gender gap as math performance increases. The authors should make explicitly clear that this is speculation, not something they directly tested. The other mechanisms were all directly tested by the authors, so my fear is that some readers may mistakenly believe stereotypes were tested as well. The authors should make clear that they could not test this but existing literature suggests it could be a possible explanation.

2) Tighten up the sentences in your manuscript and proofread carefully. Some of the sentences that were added in this latest revision feel hastily written and not well proofread. For example on p. 12 the following sentence reads: "They could possibly account for girls' and boys' differential selection into math for example because they would be more salient to students who perform well in math, for example because these stereotypes can incorporate the idea that girls cannot excel at math, a discipline where it is often believed that innate talent could be required." This is a run-on sentence that uses repetitive phrasing. The manuscript should be thoroughly combed through to improve grammar, sentence structure, and readability prior to publication.

Reviewer #3:

Remarks to the Author:

I was a reviewer on the previous two versions of this paper. Though I think the message is more focused and clear in this version, I am not convinced that the contribution of the current data is large enough to justify publication in this journal.

First, though the relationship the authors find between math performance and gender gaps in intentions is interesting, it does not seem all that different from what is already in the literature especially now that the mechanism is missing. (I support the removal of the gender stereotypes mechanism because of the measurement issues but not having anything else there also lessens the potential contribution.) To me, this paper fits into the broader knowledge that gaps are biggest among the most privileged. When it comes to performance specifically, a paper by Benbow et al in *Psych Science* (2000) that finds that high achieving girls are less likely than high achieving boys to pursue math related fields. I realized they don't have the low performing comparison like the current paper but to me adding that in is of course useful but I'm not sure it meets the bar for publication in this journal.

Second, the authors link SES to their gender stereotypes preferred mechanism, but I found this to be fuzzy and imprecise. SES might have its effects through multiple routes, as they describe (e.g., extracurriculars), but not necessarily directly through gender stereotypes. For example, perhaps higher SES children are seeing their dads work in tech and their moms work as doctors where lower SES children don't see that gendered divide to the same extent. This may be related to gender stereotypes but it is also driven by a lot of other factors, so condensing the two felt messy.

A couple of other issues came up for me in this version:

- The authors could use more care with their comparisons. When discussing the possible mechanism, they say "high performing girls could shy away more from math fields than low performing ones because of...". This is problematic because it is not true. The effect to be explained are differences in gender gaps. The authors' data shows that high performing girls are not less interested in math than low performing girls.
- The dependent measures are specific to math itself but the authors describe them as "intentions to pursue math-related studies and careers." That seems to be a leap because all of the questions are about math specifically and not about other math-related classes such as the ones mentioned in the first paragraph (e.g., computer science, physical science). The measure is also relative to

other science and reading/literature classes but that is also lost in the description and interpretation, which seems to suggest some of the science courses are captured with their measure. I'm wondering whether the contribution of this paper is high given the small number of people who go into math as a career compared to other fields such as computer science and engineering.

Answers to Reviewer #1

I have reviewed the manuscript. Overall I think the authors did a great job addressing the concerns raised by the editor and the reviewers.

Answer.

Thank you very much for your encouraging remarks

1) Regarding discussion of stereotypes as a potential mechanism that explains increasing gender gap as math performance increases. The authors should make explicitly clear that this is speculation, not something they directly tested. The other mechanisms were all directly tested by the authors, so my fear is that some readers may mistakenly believe stereotypes were tested as well. The authors should make clear that they could not test this but existing literature suggests it could be a possible explanation.

Answer:

We agree. In the revised version, to avoid confusion, we do not mention gender stereotypes in the Results section (while the other mechanisms are presented in the Results section). In fact, we refrain from any reference to stereotypes as a possible mechanism other than in the Discussion section.

Moreover, in the Discussion section, we are very cautious and make it clear that we did not test the mechanism and that it is speculation: *[...] what could drive our findings? Although we are not able to provide supporting evidence, we ultimately suggest that gender stereotypes could provide a plausible general explanation. Gender stereotypes relating math primarily with men have been considered in the literature as explanations for gender differences in choices of math/science studies and careers (23, 27-28). They could possibly account for differential selection into math among girls and boys, for example, because they would be more salient to students who perform well in math. [...] Note that gender stereotypes would then generate a self-fulfilling mechanism: the stereotype that girls cannot be excellent at math would dissuade the most talented girls from studying math, leading women to have lower math performance than men among math students, which would contribute to maintaining the initial stereotype.*

2) Tighten up the sentences in your manuscript and proofread carefully. Some of the sentences that were added in this latest revision feel hastily written and not well proofread. For example on p. 12 the following sentence reads: "They could possibly account for girls' and boys' differential selection into math for example because they would be more salient to students who perform well in math, for example because these stereotypes can incorporate the idea that girls cannot excel at math, a discipline where it is often believed that innate talent could be required." This is a run-on sentence that uses repetitive phrasing. The manuscript should be thoroughly combed through to improve grammar, sentence structure, and readability prior to publication.

Answer:

We are sorry for the sentence you quote and agree with you. Thank you very much for your advice. We followed it and had the manuscript proofread by a professional editing service.

Answers to Reviewer #3

I was a reviewer on the previous two versions of this paper. Though I think the message is more focused and clear in this version, I am not convinced that the contribution of the current data is large enough to justify publication in this journal.

First, though the relationship the authors find between math performance and gender gaps in intentions is interesting, it does not seem all that different from what is already in the literature especially now that the mechanism is missing. (I support the removal of the gender stereotypes mechanism because of the measurement issues but not having anything else there also lessens the potential contribution.) To me, this paper fits into the broader knowledge that gaps are biggest among the most privileged. When it comes to performance specifically, a paper by Benbow et al in Psych Science (2000) that finds that high achieving girls are less likely than high achieving boys to pursue math related fields. I realized they don't have the low performing comparison like the current paper but to me adding that in is of course useful but I'm not sure it meets the bar for publication in this journal.

Answer:

Let us try to address your concerns and restate our contribution.

First, we think that our findings are different from the results in the literature. As you mention, the difference with existing literature is that we analyse the gender gap in intentions to pursue math *along the ability distribution*. There is a large literature trying to explain the average gender gap, but this is not our aim. Besides, the study by Benbow et al. (2000) that you refer to does not consider either how the gender gap evolves at different points of the ability distribution. Benbow et al. (2000) consider mathematically precocious males and females tested at 13 (1,975 gifted adolescents, in the top 1% of mathematical ability) and show that twenty years later, fewer females than males were invested in the inorganic sciences and engineering. Hence, as you write in your comment, Benbow et al. (2000) show the existence of a gender gap at the very top of the ability distribution (top 1%), but do not consider the rest of the distribution. This implies that they cannot study how the gender gap evolves with students' math performance and show that this gap is higher among high math performers than among low math performers. We now make it clearer in the introduction that our analysis focuses on the whole ability distribution, unlike most previous work.

Second, we feel that you are not fully convinced by the usefulness of an analysis of the gender gap along the ability distribution and we now try to highlight more clearly and more directly in the introduction why we think that such an analysis is important: *An analysis of men's and women's choices regarding math-related studies as a function of their math ability is important for the following three main reasons. First, as mentioned, the gender gap in math-related studies is of greater concern in terms of worker shortages and wage inequality if it increases with math ability and is high among high-ability students. Second, if the gender gap in the choice to pursue math-related studies increases with math ability, that fact may imply a gender gap in math performance among math students and an underrepresentation of women among the most able students in math. Third, in terms of policy implications, an analysis along the ability distribution permits us to better identify the part of the ability distribution where interventions should be targeted to reduce the gender gap.*

In particular, concerning the second point, we show that the increase of the gender gap with math ability has implications for gender differences in math performance: the gender gap in math performance is higher among students who intend to pursue math than in the whole population, and the underrepresentation of women among the top math achievers is also higher in the former group. This means that the way in which men and women choose to pursue math has in itself implication in terms of gender differences in math performance and it is in our opinion a contribution of our work.

Third, we are sorry that you do not find the contribution sufficient. Your assessment seems driven by the fact that there is already a large literature showing that several gender gaps are larger among the “more privileged”. Let us try to restate why what we do in the present paper differs from the literature you seem to allude to. First, this literature (that you mentioned during the previous round of reviewing) is mostly based on cross-country comparisons (e.g. the papers on the “gender-equality paradox”, or the work by Maria Charles) or comparisons across school districts (e.g. Reardon et al., 2019). These papers do not attempt to relate math abilities at the individual level to educational choices, and are very different in spirit from the student-level comparisons we undertake in the present paper. Second, socioeconomic background (as a measure of “being more privileged”) and math ability are different objects. We choose to focus in this paper on math ability in part because it is policy relevant and also because it allows us to study how selection as a function of this variable can have implication for the math abilities of men and women who intend to pursue math-related studies or careers (as explained above). Additionally, socioeconomic background is only able to explain about half of the pattern we describe in the paper, suggesting that there is more than just an effect of social background in our results. Finally, we clearly situate our contribution in the line of research that evaluates how math abilities affect decisions to study math. As already explained, we make a clear contribution in that literature. The closest paper to ours is the recent study by Cimpian et al. (2020) published in *Science* that we discussed at length in the former round of revisions. To our knowledge, Cimpian et al. (2020) are the only one offering a study “along the ability distribution”. Let us remind you, however, that they reach an opposite conclusion to ours (we detail in the discussion section possible reasons for those opposite results [redacted]). This clearly shows that the fact to observe larger gender gap among the most privileged (or the most able) is not something consensual in the literature, and our paper therefore provides new evidence that challenges the current state of knowledge regarding how selection into math studies evolves with students’ math abilities.

Second, the authors link SES to their gender stereotypes preferred mechanism, but I found this to be fuzzy and imprecise. SES might have its effects through multiple routes, as they describe (e.g., extracurriculars), but not necessarily directly through gender stereotypes. For example, perhaps higher SES children are seeing their dads work in tech and their moms work as doctors where lower SES children don’t see that gendered divide to the same extent. This may be related to gender stereotypes but it is also driven by a lot of other factors, so condensing the two felt messy.

Answer:

You are perfectly right and we reorganized the text to take into account your comment. First, following the editor’s request, the results on SES are now presented in the Results section, while the interpretation for the fact that SES partly explain our main pattern is

discussed in the Discussion section. In that latter section, we mention that high SES parents (and more broadly more privileged environments) may transmit more gender stereotypes, but we are now careful to also mention other possible mechanisms, including the one you suggest here (low and high SES mothers and fathers having different occupations and acting as role models).

We think that keeping this discussion on the role of SES is useful as that is the only place in the paper where we discuss that our results could “be part of another story where gender gaps may be wider among communities that are more privileged” (we cite your comment during the previous round of revisions here), and cite the corresponding literature.

The revised discussion of SES is as follows: “*We have also shown that SES partly explains our results. This could be because high-income parents spend more time and money on their children and invest in more gendered activities (35). Hence, they may transmit gender norms, including those related to math careers (36), earlier and to a greater extent (37). At a more macro level, gender gaps in math education are known to be larger in socioeconomically advantaged school districts (37) and in richer countries (38-39). The type of educational environment that is likely to surround high math performers within a country or to prevail in more privileged school districts or in more developed countries in general typically implies more difficult curricula, higher performance standards, and greater competition. Such contexts may affect differentially girls and boys. They have in particular been shown to heighten gender essentialist ideas about math and science (40). Hence, both parental background and school environment may explain why general gender stereotypes regarding math could be more salient to high-achieving students, leading to a larger gender gap among such students. Parental background can however also affect students through other and possibly more direct channels. For example, high- and low-SES parents have different occupations on average, and perhaps the gender divide between parents’ occupations is higher among high-SES students, leading them to make more gendered educational choices. Further research would be necessary to understand exactly why SES can explain about half of the differential selection of girls and boys into math studies according to their math ability.*”

The authors could use more care with their comparisons. When discussing the possible mechanism, they say “high performing girls could shy away more from math fields than low performing ones because of...”. This is problematic because it is not true. The effect to be explained are differences in gender gaps. The authors’ data shows that high performing girls are not less interested in math than low performing girls.

Answer:

Thank you. Indeed, in the sentence you quote, it is implicit that it is 'relative to boys'. In the revised version, we are very careful and always explicitly mention that it is relative to boys when such comparisons are made.

The sentence you mention is not in the revised version because we refrain from any reference to gender stereotypes as a possible mechanism other than in the Discussion section. However, in the Conclusion for instance, we write: *Relative to their male counterparts, high-performing girls shy away from math more than low-performing girls*

The dependent measures are specific to math itself but the authors describe them as “intentions to pursue math-related studies and careers.” That seems to be a leap because all of the questions

are about math specifically and not about other math-related classes such as the ones mentioned in the first paragraph (e.g., computer science, physical science). The measure is also relative to other science and reading/literature classes but that is also lost in the description and interpretation, which seems to suggest some of the science courses are captured with their measure. I'm wondering whether the contribution of this paper is high given the small number of people who go into math as a career compared to other fields such as computer science and engineering.

Answer:

First, we adopt the same terminology as in PISA survey and reports.

We refer to our measure either as a measure of 'intentions to pursue/study math' (in the title for instance, and in many occurrences), or as a measure of 'math intentions' (to adopt a shorter formulation), or as a measure of 'intentions to pursue math-related fields or careers' (less often).

PISA survey and reports (see OECD, 2015) refer to "intentions to pursue studies or careers related to math" or to 'math intentions' when considering students' answers to the five following questions about their intention (i) to take additional math versus English/reading courses after school finishes, (ii) to take a major requiring math skills versus a major requiring science skills in college, (iii) to study harder than required in math versus English/reading courses, (iv) to take a maximum number of math versus science classes, and (v) to pursue a career that involves a lot of math versus science.

More recent papers like e.g., Gjicali-Lipnevich (2021), or Skrzypiec-Lai (2017) refer to the same measure in PISA as 'students' intentions to pursue mathematics' or 'math intentions'.

Second, we disagree that the questions are about math itself. They are not about math only, but about a major *requiring math skills* (and not only a math major), about math *vs. reading*, about a career *that involves a lot of math* (and not only a math career). This is broader than students intending to go into a career in math, as you suggest in your last sentence, and in our opinion, this is why PISA adopted the terminology 'math-related studies and careers'.

Moreover, recall that the population we consider are 15-year-old students, who predominantly are about to make their first educational choices. This is the beginning of the 'leaky pipeline' and the first educational choices they are about to make are in general, as in the questions asked in PISA survey, precisely about choosing more math-related subjects (like math or computer science) or more humanities-related subjects (like English). This is certainly the reason why items in PISA are formulated in this way. The situation would be different at college level. In our opinion, students intending to pursue a career in computer science or engineering are included in our group of 15 y.o. students intending to pursue math related studies and careers.

References:

- Benbow, C. P., Lubinski, D., Shea, D. L., & Eftekhari-Sanjani, H. (2000). Sex differences in mathematical reasoning ability at age 13: Their status 20 years later. *Psychological Science*, 11(6), 474–480. <https://doi.org/10.1111/1467-9280.00291>
- GjicaliK., Lipnevich, A. (2021) Got math attitude? (In)direct effects of student mathematics attitudes on intentions, behavioral engagement, and mathematics performance in the U.S. PISA. *Contemporary Educational Psychology* 67 (2021)

- Skrzypiec, G. and Lai, M. (2017) Social Psychology Meets School Mathematics in PISA 2012: An Application of the Theory of Planned Behaviour in Australia. *Psychology*, **8**, 2146-2173. doi: [10.4236/psych.2017.813137](https://doi.org/10.4236/psych.2017.813137).

Reviewers' Comments:

Reviewer #1:

Remarks to the Author:

I have carefully reviewed the author's responses to the comments by the reviewers. Overall, I believe the authors did a good job addressing the reviewers' comments. I have one minor issue that the authors should correct: On p. 2, line 39, the authors state that women are not underrepresented in certain STEM fields, such as the life sciences and psychology. I have never heard of psychology being identified as a STEM field. Psychology is a social science, which is one of the reasons why it has such a high representation among women. The social science fields are overwhelmingly female, so I believe this statement is false and misleading. I would suggest that the authors give specific examples of STEM fields where women are highly represented, such as biology or medicine and maybe present percentages to back it up. Other than that, I think the authors were very responsive to comments.

Reviewer #3:

Remarks to the Author:

I have now reviewed three versions of this manuscript and have been gratified to see the immense improvement with each version including in message, clarity, and scope. I think this paper is ready for publication in Nature Communication.

My final comments are small in case they are helpful to the authors:

- I was surprised to see the Discussion not to a summary of results and go straight to limitations. A summary would be helpful for the reader to process the results before heading into limitations.
- I wondered if the SES explanation would make sense to go before the gender stereotypes explanation, given the greater empirical support. It's also unlikely that the gender stereotypes explanation contributes to explaining more than half the effect.

Point-by-point answers to Reviewers

Point-by-point answers to Reviewer #1 (Remarks to the Author):

I have carefully reviewed the author's responses to the comments by the reviewers. Overall, I believe the authors did a good job addressing the reviewers' comments. I have one minor issue that the authors should correct: On p. 2, line 39, the authors state that women are not underrepresented in certain STEM fields, such as the life sciences and psychology. I have never heard of psychology being identified as a STEM field. Psychology is a social science, which is one of the reasons why it has such a high representation among women. The social science fields are overwhelmingly female, so I believe this statement is false and misleading. I would suggest that the authors give specific examples of STEM fields where women are highly represented, such as biology or medicine and maybe present percentages to back it up. Other than that, I think the authors were very responsive to comments.

Thank you very much for your positive assessment. Regarding psychology being within STEM, there is actually no clear consensus. There is indeed no universal agreement on which disciplines are included in STEM; in particular whether or not the science in STEM includes social sciences, such as psychology, sociology, economics, and political science. In the United States, these are typically included by organizations such as the National Science Foundation and many organizations in the United States follow the guidelines of the National Science Foundation on what constitutes a STEM field. To avoid being misleading, we have corrected the text and we now provide specific examples for the U.S. that will for sure not generate any confusion. We write: “STEM is however a broad group that includes fields in which women are not anymore underrepresented, such as molecular biology and neuroscience in the United-States.”

Point-by-point answers to Reviewer #3 (Remarks to the Author):

I have now reviewed three versions of this manuscript and have been gratified to see the immense improvement with each version including in message, clarity, and scope. I think this paper is ready for publication in Nature Communication.

Thank you very much for your positive assessment.

My final comments are small in case they are helpful to the authors:
- I was surprised to see the Discussion not to a summary of results and go straight to limitations. A summary would be helpful for the reader to process the results before heading into limitations.

We have added such a summary at the beginning of the Discussion. See on p. 10:
“We have shown that the gender gap in intentions to pursue math increases with students’ math performance. Relative to their male counterparts, high-performing females intend less to pursue math than low-performing females. This pattern implies that among students intending to pursue math, gender gaps in math performance get larger than in the general population. It cannot be explained by gender gaps in comparative advantage for reading

relative to math, nor by gender gaps in math self-concept or interest for math. It is however partly explained by students' SES. In this section, we discuss some limitations of our analysis as well as how these results fit into the literature."

- I wondered if the SES explanation would make sense to go before the gender stereotypes explanation, given the greater empirical support. It's also unlikely that the gender stereotypes explanation contributes to explaining more than half the effect.

This is an excellent suggestion. We have followed it and now write (p. 12):

"A first limitation is that we cannot provide evidence for a clear mechanism that explains our results. There are indeed various factors that could be behind the differential relationship between math performance and males' and females' intentions to pursue math. One of these factors is SES that can explain about half of the differential relationship. This could be for several reasons. First, it could be because high-income parents spend more time and money on their children and invest in more gendered activities (27). Hence, they may transmit gender norms, including those related to math careers (28), earlier and to a greater extent (29). At a more macro level, gender gaps in math education are known to be larger in socioeconomically advantaged school districts (29) and in richer countries (30-31). The type of educational environment that is likely to surround high math performers within a country or to prevail in more privileged school districts or in more developed countries in general typically implies more difficult curricula, higher performance standards, and greater competition. Such contexts may affect differentially females and males. They have in particular been shown to heighten gender essentialist ideas about math and science (32). Hence, both parental background and school environment may explain why general gender stereotypes regarding math could be more salient to high-achieving students, leading to a larger gender gap among such students. Parental background can however also affect students through other and possibly more direct channels. For example, high- and low-SES parents have different occupations on average, and perhaps the gender divide between parents' occupations is higher among high-SES students, leading them to make more gendered educational choices. Further research would be necessary to understand exactly why SES can explain about half of the differential relation between females' and males' intentions to pursue math and their math ability.

In all cases, SES can only explain half of our main result. What could drive the unexplained part? Although we are not able to provide supporting evidence, we ultimately suggest that gender stereotypes could provide a plausible general explanation. Gender stereotypes relating math primarily with males have been considered in the literature as explanations for gender differences in choices of math and science studies and careers (23, 33-34). They could possibly account for the differential relation between intention to pursue math and math performance among females and males, for example, because they would be more salient to students who perform well in math. Indeed, these stereotypes can incorporate the idea that females cannot excel at math, a discipline where it is often believed that innate talent is needed. The existence of the stereotype that females cannot excel at math has been documented in the literature (2, 35-36). This could imply that when facing different possible curricula, high-performing females may stay away from math and assume that they have a better chance to realize their full potential if they specialize in another field. This interpretation is also supported by several papers that have

documented that "it is often the most talented members of stereotyped groups that are most affected by others' biased perceptions and, more generally, by signals suggesting that they may lack ability" (37; see also 38-40). Note that gender stereotypes would then generate a self-fulfilling mechanism: the stereotype that females cannot be excellent at math would dissuade the most talented females from studying math, leading females to have lower math performance than males among math students, which would contribute to maintaining the initial stereotype."